# Regulation of the cell division hydrolase RipC by the FtsEX system in *Mycobacterium tuberculosis*

Jianwei Li [1,5], Xin Xu [1,5], Jian Shi[2], Juan A. Hermoso [3] ✉, Lok-To Sham [4] ✉ & Min Luo [1,2] ✉

The FtsEX complex regulates, directly or via a protein mediator depending on bacterial genera, peptidoglycan degradation for cell division. In mycobacteria and Gram-positive bacteria, the FtsEX system directly activates peptidoglycan-hydrolases by a mechanism that remains unclear. Here we report our investigation of *Mycobacterium tuberculosis* FtsEX as a non-canonical regulator with high basal ATPase activity. The cryo-EM structures of the FtsEX system alone and in complex with RipC, as well as the ATP-activated state, unveil detailed information on the signal transduction mechanism, leading to the activation of RipC. Our findings indicate that RipC is recognized through a "Match and Fit" mechanism, resulting in an asymmetric rearrangement of the extracellular domains of FtsX and a unique inclined binding mode of RipC. This study provides insights into the molecular mechanisms of FtsEX and RipC regulation in the context of a critical human pathogen, guiding the design of drugs targeting peptidoglycan remodeling.

Most prokaryotic cells are surrounded by a peptidoglycan (PG) layer composed of glycan strands of N-acetyl glucosamine and N-acetyl muramic acid connected by stem peptide crosslinks, providing mechanical support against osmotic rupture and maintaining the cell shape[1-3]. To grow the cell envelope, synthesis and hydrolysis of PG must be carefully coordinated to avoid cell lysis. The PG hydrolases for remodeling are typically recruited and organized into one of the two supramolecular complexes called the divisome and the elongasome, which contain partner PG synthases and regulators. Otherwise, in isolation, the activities of these PG hydrolases are usually autoinhibited[1-3].

The FtsEX complex is a widely conserved regulator of peptidoglycan (PG) hydrolases. It belongs to the type VII subfamily of ABC transporters and is similar to MacB and LolCDE[4-6]. The substrates of the LolCDE complex are lipoproteins, whereas the MacB complex is involved in the transport of antibiotics and virulence factors.

Conversely, as of the current understanding, the FtsEX complex does not associate with a known substrate. Instead, it functions as a transmembrane signal transduction system for activating PG hydrolases. Briefly, the ATPase activity by the cytoplasmic FtsE powers the system through its ATP cycle, interacting with the transmembrane FtsX. The signal is then transmitted from the cytosolic FtsE to the two extracellular loops of FtsX via conformational changes[7,8], and finally to the coiled-coil (CC) domain proteins. The CC domain protein can be another regulator that bridges the interaction between FtsEX and the hydrolases, such as the EnvC-AmiB system in *Escherichia coli*[9]. Alternatively, the CC domain protein can be the hydrolase itself, exemplified by PcsB[10,11], CwlO[12,13], and RipC (Rv2190c/Cg2401)[14,15]. Here, we are following the naming system of Rip hydrolases based on the phylogenetic analysis by Gaday et al[16]. RipA (Rv1477/Cg1735 or "RipC" in (14)) interacts with FtsEX and another activator SteAB in *Corynebacterium glutamicum*[13]. Consistently, deletion mutants of these genes

[1]Department of Biological Sciences, Faculty of Science, National University of Singapore, Singapore, Singapore. [2]Center for Bioimaging Sciences, Department of Biological Sciences, National University of Singapore, Singapore, Singapore. [3]Department of Crystallography and Structural Biology, Instituto de Química-Física "Blas Cabrera", Consejo Superior de Investigaciones Científicas, Madrid, Spain. [4]Infectious Diseases Translational Research Programme and Department of Microbiology and Immunology, Yong Loo Lin School of Medicine, National University of Singapore, Singapore, Singapore. [5]These authors contributed equally: Jianwei Li, Xin Xu. ✉e-mail: xjuan@iqfr.csic.es; lsham@nus.edu.sg; dbslmin@nus.edu.sg

phenocopied each other, suggesting that they are in the same pathway[14,16]. Corroborating the genetic evidence, purified RipA (Cg1735) can be activated directly by the extracellular domain of SteB[16]. Besides FtsEX and SteB, RipA can also be regulated by proteolysis or other factors[17]. In *Mycobacterium tuberculosis*, RipC (Rv2190c/Cg2401), but not RipA (Rv1477/Cg1735), is shown to be the partner CC domain PG hydrolase that interacts with FtsEX[15]. However, the mechanism by which mycobacterial FtsEX (*Mtb*FtsEX) activates PG hydrolysis is poorly understood as there is no structural information on the full-length FtsEX complex in the presence or absence of RipC.

There are several extraordinary features of cell division in *Corynebacterineae* that distinguish them from classical rod-shaped model bacteria. First, septa placement is relatively irregular, leading to a wide range of daughter cell sizes[18,19]. In addition, daughter cells are separated by a "V-snapping" mechanism[20,21] presumably due to the uneven cleavage of the stress-bearing PG layer. Deletion of FtsEX and the associated hydrolase also delays V-snapping in *C. glutamicum*[14], although how they achieve uneven cleavage of PG is still unclear. Lastly, the composition of the mycobacterial divisome is notably different[22,23]. It lacks FtsA and FtsN but instead, it has a few species-specific proteins like SepIVA, Wag31, FhaB, CrgA, and CwsA[24,25]. These proteins perhaps functionally replace the canonical division proteins found in other bacteria[22,23]. In addition, it is supposed that the unique interactions observed between cell division proteins in mycobacteria explain the absence of factors like FtsA[26]. Because of this, it could be expected that the mycobacterial FtsEX presents distinct features compared with other bacterial species.

Here, we determined the structure of *Mtb*FtsEX (FtsE: Rv3102c, FtsX: Rv3101c) complex with or without RipC (Rc2190c/Cg2401) and ATP using cryo-EM. Our biochemical investigations revealed that *Mtb*FtsEX has a high basal ATPase activity. ATP binding is not required for RipC recruitment, but it seems to affect the activation of the complex. We discovered a unique inclined binding mode of RipC that allows for the hydrolase to stretch into the extracellular space for about ~8 nm. We show that RipC recruitment is driven by the highly dynamic features of the hinge region of the connecting transmembrane helices of FtsX and a "Phe-cluster" present at the lower lobe of the extracellular domain (ECD). Additionally, we elucidated a series of conformational changes in the mechanotransmission from cytosolic FtsE ATP binding to the extracellular activation of RipC. Our results provide valuable insights into the molecular mechanisms of FtsEX and its regulation of RipC, as well as an approach to the molecular basis of the "V-snapping" cell division mechanism in *Mycobacterium*.

## Results

### Biochemical reconstitution and characterization of the FtsEX system in *Mycobacterium tuberculosis*

The full-length FtsE and FtsX proteins from *Mycobacterium tuberculosis* (str. ATCC 35743) were overexpressed in *E. coli* strain BL21(DE3), and purified using affinity and size-exclusion chromatography (Supplementary Fig. 1a). FtsEX was extracted in DDM and reconstituted in peptidiscs for functional and structural studies (see Methods). ATPase activity assays revealed that unlike other reported FtsEX complexes[8], the *Mtb*FtsEX complex presents a significant basal ATPase activity (Fig. 1a). In order to disclose the importance of ATP binding and ATP hydrolysis in the whole mechanism, we constructed two FtsE mutants based on related ABC transporters[27] that were unable to bind (D164A) or to hydrolyze (E165Q) ATP. As expected, in both cases the ATPase activity was drastically reduced (Fig. 1a).

To investigate the role of ATP binding/hydrolysis on RipC recruitment, a protein pull-down assay was performed by mixing purified FtsEX and RipC in the presence of ATP and various ATP analogs (Fig. 1b). We found that RipC forms a stable complex with FtsEX and its binding was not dependent on ATP. We also investigated the

effect of RipC binding on the ATPase activity of FtsEX (Fig. 1c) and observed no change in activity, even with a RipC concentration ten times higher than that of FtsEX. Furthermore, we found that the ATPase activity with different ATP concentrations followed a similar Michaelis-Menten curve regardless of the presence of RipC (Fig. 1d), which indicates that RipC binding did not alter ATP affinity. The Michaelis constant of *Mtb*FtsEX was ~0.2 ± 0.04 mM and slightly decreased to about 0.16 ± 0.04 mM in the presence of RipC. The catalytic rate of *Mtb*FtsEX was around 58 ± 2.9 nmol/mg/min and reduced to about 55 ± 3.7 nmol/mg/min in the presence of RipC. These results confirmed that RipC did not affect the ATPase kinetics of *Mtb*FtsEX. Although *Mtb*FtsEX had a high affinity for ATP compared to other typical ABC transporters, its catalytic rate was not as high.

### Structure of *Mtb*FtsEX reveals unusual conformation for the ECD domains

Single-particle Cryo-EM analysis allowed structural determination of the complete WT FtsEX (Fig. 1e, f and Supplementary Fig. 1b–h) with an overall resolution of 3.9 Å. *Mtb*FtsEX differs from other Type VII ABC transporters because it only has a single coupling helix (CH) (Fig. 1e, f). Cryo-EM density provides a perfect description of the full oligomeric arrangement for the FtsEX system (Fig. 1g and Supplementary Fig. 2) composed by two chains of the cytosolic FtsE NBD domains and two chains of the FtsX with TMD domains inserted into the membrane and the ECD domains protruding to the periplasmic space. The TMD and NBD regions of *Mtb*FtsEX structure are similar to other Type VII ABC transporters, such as MacB[28], LolCDE[29] and *Pae*FtsEX[8]. The TMD domain contains four long transmembrane helices (TM1-4) in close contact inside the dimer (Fig. 1h). The N-terminal region of FtsX (residues 1-17) presents an elbow helix (EH) parallel to the membrane. Dimerization of the two FtsX protomers occurred through extensive hydrophobic interactions between TM2 and TM3, and the interface between FtsE and FtsX involved a large interacting area of ~960 Å², contributing to a stable *Mtb*FtsEX complex with a robust ATPase activity.

The ECD domains in both FtsX monomers presented a well-defined density. The small loop (SL) comprises eight residues between TM3 and TM4 (residues 258–265) (Fig. 1e, f). The ECD spans residues between TM1 and TM2 (residues 54-156) and adopts a canonical topology (β2αβ3αβαβ). The two sets of ECD domains are structurally superimposable to each other (Supplementary Fig. 3a), and they closely resemble the previously determined crystal structure *of Mtb*ECD[15] (Supplementary Fig. 3b). Each ECD domain is divided into two lobes: the upper lobe, which includes β1, α1, α2, β2, β3, α6, and β4, and the lower lobe, which consists of α3-α5. Structures of the ECD in Type VII family members have all shown an upper lobe and a lower lobe, which together form a hook-shaped structure that is crucial for recruiting substrate ligands and interacting proteins. Through our comparison of the ECD structures, we have found that the upper lobe of FtsX shares a similar fold with those of other Type VII transporters, such as MacB and LolCDE (Supplementary Fig. 3c)[28,29]. However, the lower lobe of the ECD of FtsX is notably smaller in size in Mycobacterium[15] (and this work), and also in *Pseudomonas aeruginosa*[8] and in *Streptococcus pneumoniae*[10]. Besides the *Mtb*FtsEX structures in this study, *Pae*FtsEX from a Gram-negative bacteria of *P. aeruginosa* is the only FtsEX complex with a fully resolved structure to date[8]. Although *Mtb*FtsEX and *Pae*FtsEX have only 18% sequence identity, the overall structure of their ECD domains is quite similar with an RMSD of 1.16 Å for Cα atoms. Interestingly, the ECD of *Mtb*FtsEX contains a short α helix (residues 63-69) and a disulfide bridge, which is not present in other ECDs of FtsX (Supplementary Fig. 3d).

Remarkably, a significant conformational difference becomes apparent when comparing the dimeric arrangement of *Mtb*FtsEX with other VII ABC transporters (Fig. 2). Specifically, while the ECD dimer of other Type VII ABC transporters, such as MacB[28], LolC[29], and the

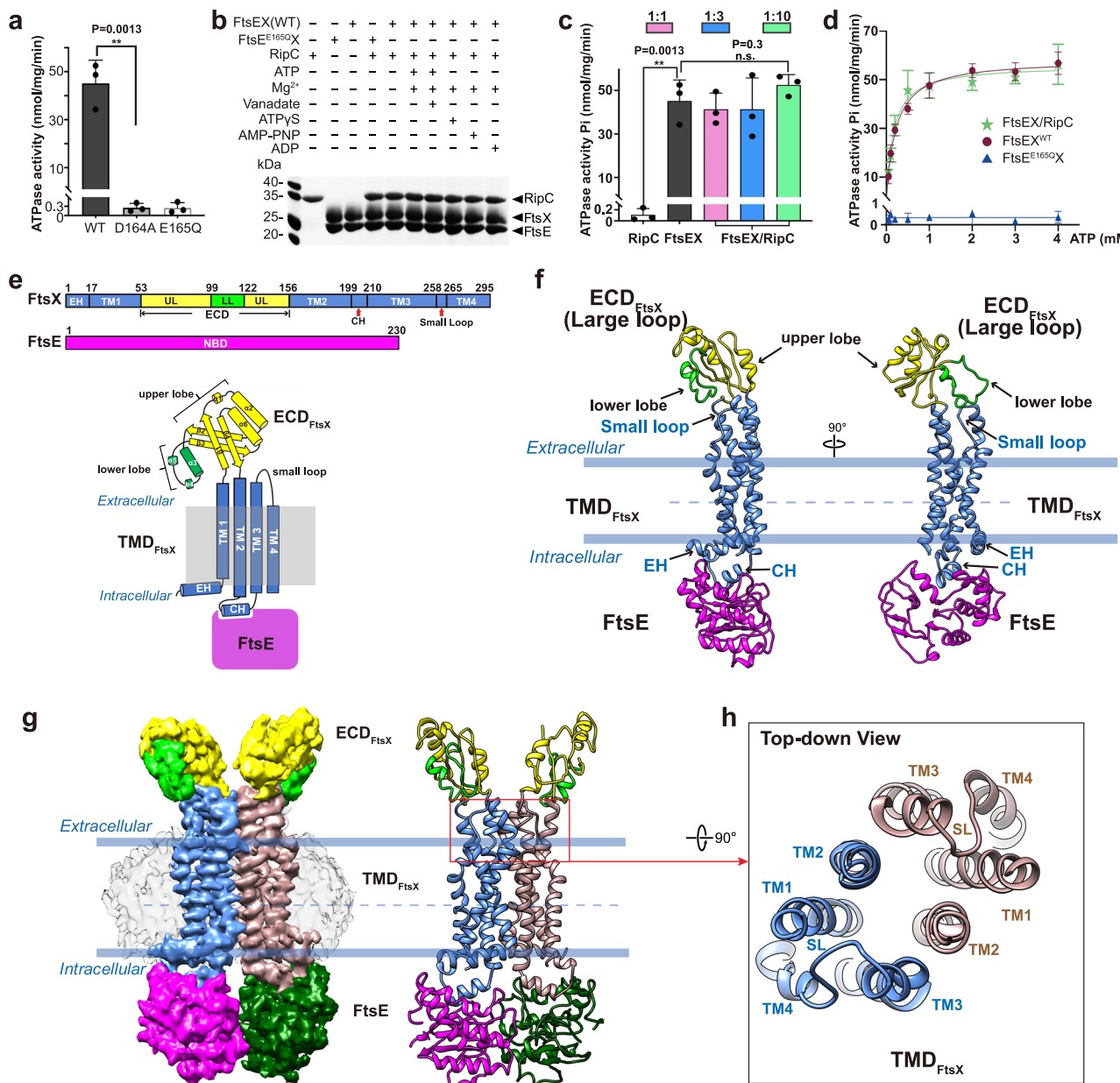

**Fig. 1 | Biochemical and structural study of *Mtb*FtsEX complex in peptidisc.**
**a** ATPase activity of FtsEX and FtsEX mutants. There are no ATPase activity of mutants FtsE$^{D164A}$X and FtsE$^{E165Q}$X. $n = 3$ replicates, error bars were presented as mean ± SD. Two-tailed unpaired $t$-test; **$P < 0.005$. The $P$-value is shown in figure. **b** Pull-down study of RipC binding to FtsEX or its ATPase mutant, in the presence or absence of different nucleotides or analogues. RipC forms a stable complex with FtsEX was ATP independent. Each experiment was repeated three times independently with similar results. **c** ATPase activity of FtsEX in the presence of RipC. The addition of RipC has no effect on the ATPase activity of FtsEX. $n = 3$ replicates, error bars were presented as mean ± SD.Two-tailed unpaired $t$-test; **$P < 0.005$, n.s. no significant difference. The $P$-values are shown in figure. **d** ATPase activity of FtsEX and FtsEX/RipC complex in peptidisc. $n = 3$ replicates, error bars were presented as

mean ± SD. The source data of **a**–**d** are provided as a Source Data file. **e** Domain arrangements and topology diagram of FtsX and FtsE. For FtsX, α helices are shown as cylinder, β sheets are shown as arrows. **f** Protomer structure of FtsEX in 2 orientations. Color scheme: FtsE in magenta, TM$_{FtsX}$1-4, CH$_{FtsX}$ and EH$_{FtsX}$ in cornflower blue, the upper lobe of ECD$_{FtsX}$ in yellow, and the lower lobe of ECD$_{FtsX}$ in light green. CH coupling helix, EH elbow helix, UL upper lobe, LL lower lobe. **g** Front-view of the cryo-EM density map of FtsEX in the absence of ATP (left), and the ribbon representation of WT FtsEX (right). Color scheme: FtsX in blue and brown, The upper lobe of ECD$_{FtsX}$ in yellow, and the lower lobe of ECD$_{FtsX}$ in light green; FtsE in magenta and dark green. **h** Top-down view of TMDs with small loop (SL) shown.

recently resolved *Pae*FtsEX[8], are oriented with their lower lobes facing each other (Fig. 2a, b), the two ECD domains of *Mtb*FtsEX face aside in opposite directions. To understand the structural differences, we compared their structures and found an obvious kink in TM1 of *Mtb*FtsEX, which is absent in other Type VII structures (Fig. 2c). The kink occurs at the hinge region of TM1 that directly connects to the ECD domain. In *Aa*MacB and *Ec*LoLC, a straight conformation is adopted, making it parallel to another hinge region of TM2 that is

connected to the ECD domain from the other side. The kink in TM1 of *Mtb*FtsEX alters the orientation of the coupled ECD domain, resulting in a distinct conformation in which the two ECD hooks with the two lower lobes facing in opposite directions. These lower lobes play an essential role in binding cognate partners as observed in the *Ec*FtsX-PLD domain[30]. This unique arrangement leaves a wide-open groove between them, raising the question of how it can bind its cognate hydrolase RipC for PG hydrolysis.

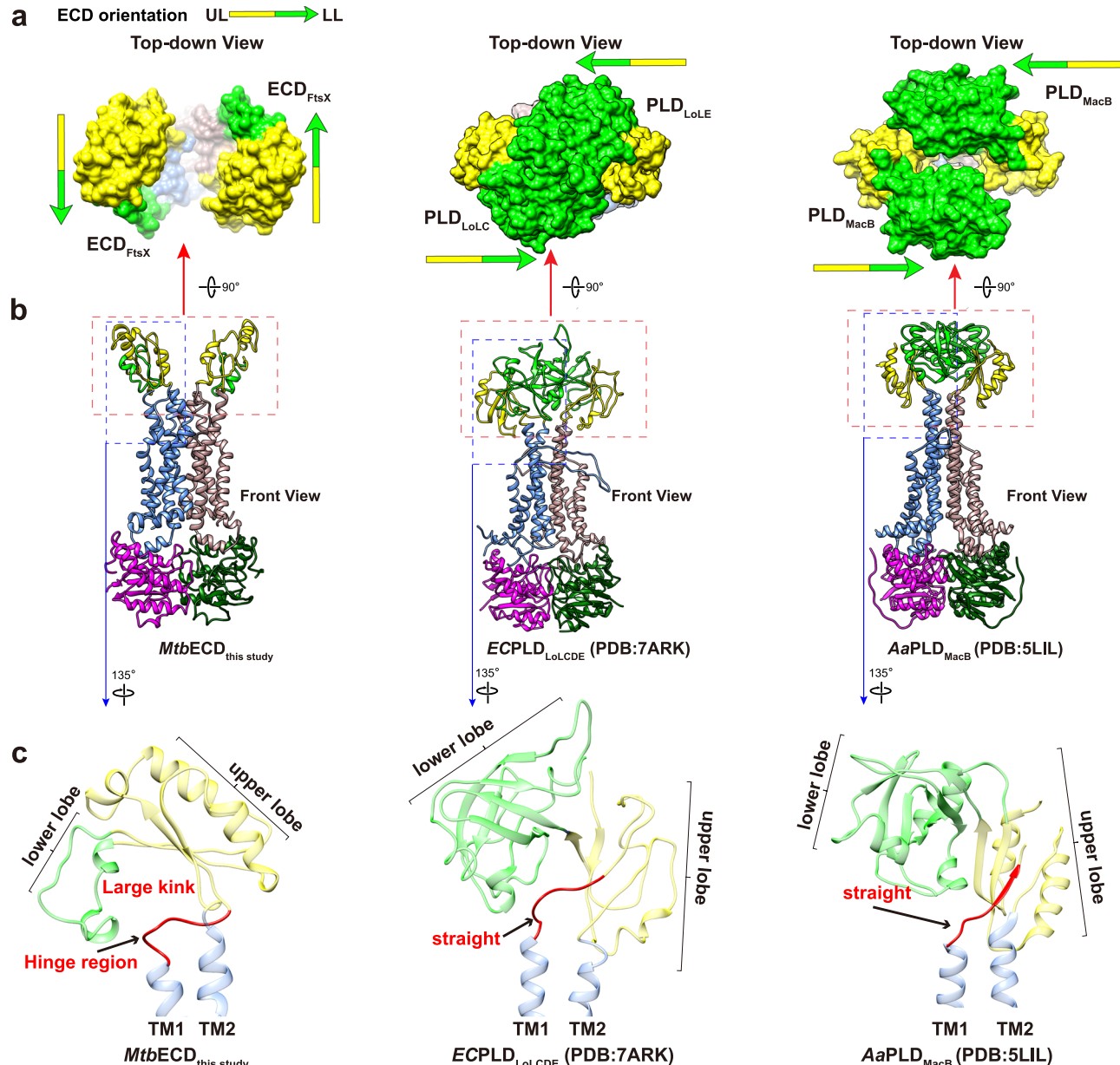

**Fig. 2 | *Mtb*FtsEX exhibits unusual ECD domain conformation compared to other Type VII ABC transporters. a** The surface representations of the ECD/PLD domains are shown (top-down view), with the upper lobe of the ECD/PLD colored yellow and the lower lobe colored in light green. In *Mtb*FtsEX, the two lower lobes of the ECD face in opposite directions, whereas in *Aa*MacB (PDB code: 5LIL) and *Ec*LolC (PDB code: 7ARK), the two lobes face to each other. UL upper lobe, LL lower lobe. **b** Structural comparison of ECD/PLD domain in three Type VII ABC transporters with structures available: *Mtb*FtsEX, *Aa*MacB, and *Ec*LolC. The structures are represented by ribbon (front view). **c** Structural comparison of an ECD/PLD monomer, along with the connecting hinge region, between *Mtb*FtsEX, *Aa*MacB, and *Ec*LolC. The hinge region from TM1 in *Mtb*FtsEX exhibits a large kink conformation, while the corresponding regions in *Aa*MacB and *Ec*LolC maintain a straight conformation, leading to a different orientation of the connecting ECD monomer. The hinge region from TM1 is highlighted in red.

## Cryo-EM structure of FtsEX-RipC complex in ATP-free state, revealing an autoinhibited structure of RipC with Catalytic site blocked by α1

Next, we used cryo-EM to determine the structure of the FtsEX-RipC complex (Supplementary Fig. 4). The final map achieved a resolution of 3.9 Å (Supplementary Fig. 5). This allowed us to construct a complete model of the complex between FtsE, FtsX, and RipC. The structure of RipC is here presented containing all amino acid residues except part of the proline-rich linker region (Fig. 3a). It consists of a signal peptide (residues 1-44), a coiled-coil domain (CCD) at the N-terminus (residues 45-216), a proline-rich linker region of 53 residues (residues 217-269), and a conserved NlpC/P60 family catalytic domain (residues 270-385) at the C-terminus (Fig. 3a). The proline-rich linker

includes 18 Pro, 11 Ala, 9 Gly, 9 hydrophobic residues and just only 4 polar residues; of these, 12 amino acids (217-228) were visible in the density and the remaining were disordered (Supplementary Fig. 6) The CCD includes two long α-helices, α1, α2, and a lip (linear interacting peptide) region in between. The lip region, consisting of 18 residues (residues110-128), forms one end of RipC that links α1 and α2; on its distal end is the NlpC/P60 catalytic domain. Our captured RipC is in an autoinhibited state, as the N-terminus of CCD-α1 restrains the conserved catalytic site (Fig. 3a).

The structure of the catalytic domain of RipC lacks the self-inhibitory helix or polypeptide that typically occludes the catalytic domain of other cell-division hydrolases such as AmiB[31], AmiC[32] or RipA[33]. Instead, in RipC, α1 serves as the regulatory helix and is closer

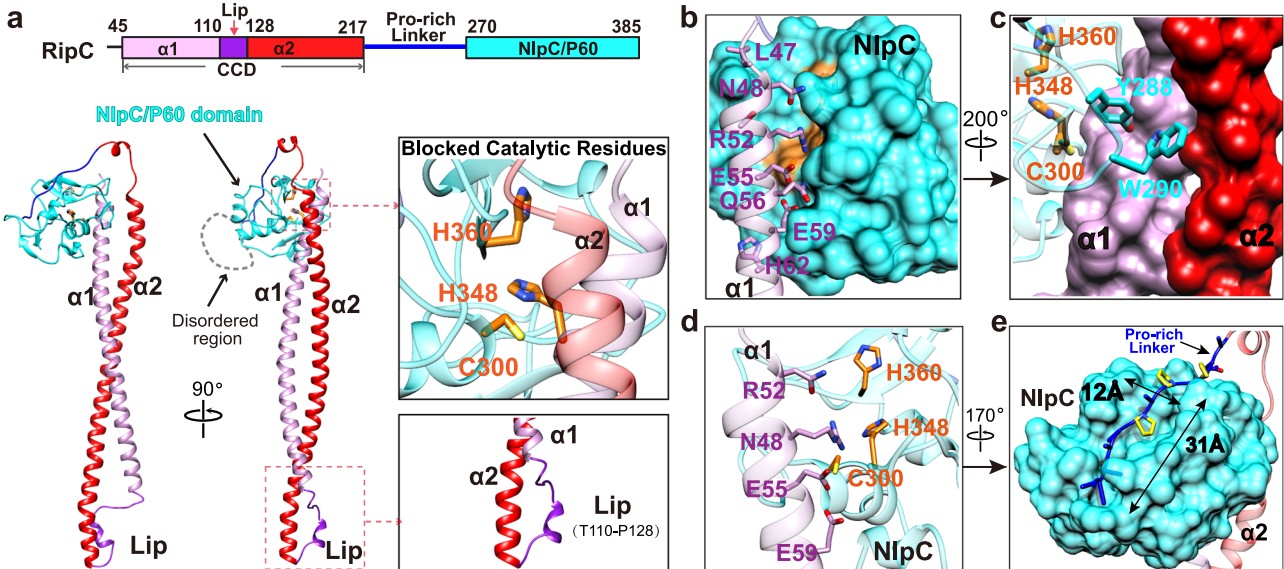

**Fig. 3 | The structure of RipC. a** Domain arrangements and overall structure of RipC in an autoinhibited state. The α1 domain is colored in plum, lip in purple, α2 in red, pro-rich linker in blue, the NlpC/P60 catalytic domain in cyan, and the catalytic residues are in orange. **b** The NlpC/P60 domain is restrained by the N-terminal region of α1. The NlpC/P60 domain is shown in cyan surface, while the residues involved in the interaction with α1 are depicted in plum sticks. **c** Y288 and W290 of the NlpC/P60 domain (in cyan) are positioned like a wedge, inserted into a shallow groove between α1 and α2 helices. **d** Interaction between the NlpC/P60 domain and α1 helix blocks access to the catalytic site of NlpC/P60. Catalytic residues are shown in orange sticks. **e** The initial segment of the Pro-rich linker connecting the CCD with the catalytic domain is stabilized in a long groove of the catalytic domain. Catalytic domain is represented in surface, the CCD is shown in ribbon with 60% transparency.

to the PG hydrolase Cg1735 (the RipA homologue in *Corynebacterium glutamicum*) in which the catalytic domain is obtruded by its N-terminal conserved coiled-coil domain[16] or to the pneumococcal PcsB hydrolase in which the CCD resembles molecular tweezers locking the PcsB's catalytic domain in an inactive state[34]. RipC presents the canonical catalytic triad Cys-His-His (C300, H348, H360 respectively) (Fig. 3a and Supplementary Fig. 7a, b) observed in most of NlpC/P60 domains[35]. Interestingly, this configuration is not observed in RipA and its homologs (RipB, Cg1735) that preserve the first two catalytic residues, but present a Glu residue completing the catalytic triad (Supplementary Fig. 7c–h). Other relevant differences between RipC and RipA homologs are also observed in the loops around the catalytic center (Supplementary Fig. 7i) that result in a different shape and dimensions for the active site (Supplementary Fig. 7j).

In the autoinhibited state, the NlpC/P60 domain of RipC is stabilized by extensive interactions with the N-terminus of CCD-α1 (residues 47-62) (Fig. 3b), with bulky amino acids like Y288 and W290 acting as a wedge and inserted into the shallow groove region between α1 and α2 (Fig. 3c). In addition, a cluster of polar residues from the N-terminal of α1, including N48, R52, E55, and E59, are located within hydrogen bond range with the catalytic site of NlpC/P60, locking the catalytic residues of C300, H360, and H348 underneath (Fig. 3d) similarly to that observed in Cg1735 from *C. glutamicum*[16] (Supplementary Fig. 7 h). Interestingly, the beginning of the linker region after CCD-α2 is strongly stabilized by hydrophobic interactions within a large and deep groove (~31 Å x 12 Å x 8 Å) of the NlpC/P60 domain located at the back of the active site (Fig. 3e). Thus, the linker region and the CCD hold the hydrolase domain in an inactive state resembling the molecular tweezers observed in pneumococcal PcsB[34]. These results suggest that RipC activation is required to initiate PG hydrolysis, and it appears that activation could be achieved by moving the blocking α1 helix away from the NlpC/P60 domain.

## Molecular recognition of RipC by FtsEX

Our structure of the complete FtsEX-RipC complex (Fig. 4a) reveals a well-resolved binding interface between FtsEX and RipC. The lip of

RipC, as well as its adjacent α1 region (residues 102-109) and α2 region (residues 129-148), bind directly to the ECD of FtsX, resulting in a buried surface area of ~1850 Å². This binding involves three major interfaces that are dominated by extensive hydrophobic interactions, two of which involve a Phe cluster consisting of four phenylalanine residues (F61, F110, F113, and F122) from each of the two lower lobes of the ECD domains (Fig. 4b, c). These Phe residues are tightly packed against RipC, with Y111 on the complementary side of α1$_{RipC}$ engaging in multiple π−π interactions with the Phe cluster, and its neighboring M112 forming a strong methylene-aromatic interaction with F61. M141 from α2$_{RipC}$ is involved in a methylene-aromatic interaction with the Phe cluster from another ECD. Notably, a cluster of hydrophobic residues from the lower lobe of FtsX, including W123 and F126, were also shown to be essential in the interaction with PcsB in the pneumococcal case[10]. The third interface involves the top surface region of the transmembrane helices with the lip of RipC and mainly involves hydrophobic interactions, including residues Y52 or I122. (Fig. 4d). In addition, a few hydrogen bonds, such as R115$_{RipC}$-N258$_{FtsX}$ and Q129$_{RipC}$-R55$_{FtsX}$, are also present. Several electrophilic residues, such as R161 from FtsX, are within hydrogen bond range of the lip but do not form significant electrostatic interactions. In vitro pull-down assays (Fig. 4e) further revealed that mutation of the Phe cluster (F61, F110, F113, and F122) and Y52 in FtsX, or their interacting residues Y111 and M112 in RipC, abolished the interaction between the two proteins. However, disrupting the hydrogen bonds by mutating N258, R55, and R161 in FtsX, or Q129 and R115 in RipC, did not affect the interaction. As anticipated, mutating these residues had no discernible effect on the ATPase activity (Supplementary Fig. 8), since RipC binding does not directly modulate ATP hydrolysis. These findings suggest that the Phe cluster in FtsX and the extensive hydrophobic interactions, rather than charged residues, play a key role in binding to RipC. The overall binding pocket for RipC resembles a bone joint cavity that locks the bulky region from the end of the lip side inside, allowing the distal NlpC/P60 domain end of RipC to protrude into the extracellular space with a tilt toward the PG layer.

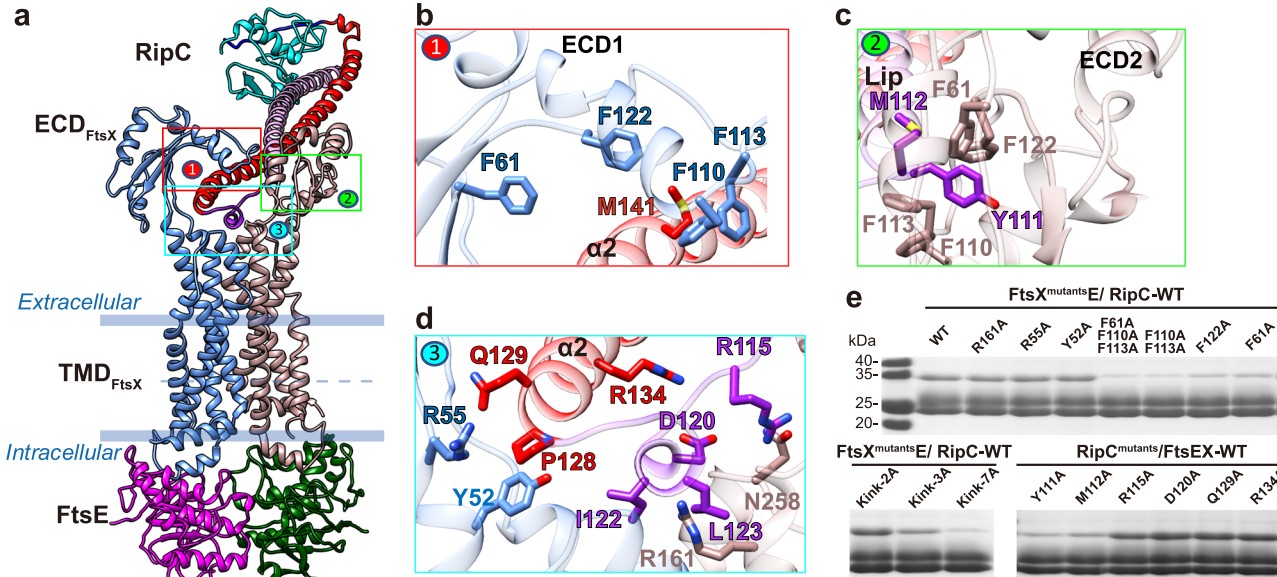

**Fig. 4 | The detailed interactions between *Mtb*FtsEX and RipC. a** The RipC is bound to the ECD domain of FtsEX. The structures are represented by ribbon. Color scheme: FtsX is indicated in cornflower blue/rosy brown, FtsE is indicated in magenta/dark green. For RipC, α1 in plum, lip in purple, α2 in red, pro-rich linker in blue, the NlpC/P60 catalytic domain in cyan. **b–d** Interaction details (front view) of interface 1, 2 and 3 between FtsX and RipC. Critical residues are shown with sticks.

**e** RipC binding assay of FtsX and RipC mutants from FtsX-RipC interface. In shorts, Kink-2A: the ECD_FtsX kink region mutant D54A-R55A. Kink-3A: the ECD_FtsX kink region mutant I51A-Y52A-L53A. kink-7A: the ECD_FtsX kink region mutant R49A-A50A-I51A-Y52A-L53A-D54A-R55A. Each experiment was repeated three times independently with similar results. Source data are provided as a Source Data file.

## A unique inclined binding mode of RipC

Strikingly, the FtsEX-RipC complex differs significantly from that of EnvC-bound *Pae*FtsEX (PDB code: 8I6O)[8] (Fig. 5a). While both binding partners are anchored in between the two ECD domains, the CC domain of EnvC aligns roughly with the central axis of FtsEX, resulting in an elongated conformation. In contrast, RipC is inserted into the ECD domain from the side at a ~116° tilt from the central axis, resulting in an inclined binding mode. Despite the tertiary structures of the ECD monomers being highly superimposable, the binding modes of their putative partners are drastically different, leaving the molecular details underlying interesting.

To understand the differences in binding modes, we compared the structures of RipC-bound *Mtb*FtsEX and EnvC-bound *Pae*FtsEX structures (PDB code: 8I6O) (Fig. 5b). In EnvC-bound *Pae*FtsEX, the two ECD domains are facing each other. In RipC-bound *Mtb*FtsEX, one ECD monomer faces the central axis due to the straight hinge conformation from TM1 and TM2, while the other ECD monomer has a kink in the hinge of TM1, causing the lower lobe of the ECD domain to face away from the central axis at an angle of ~90°. As a result, in *Pae*FtsEX, EnvC slices in between the two parallel ECD-hooks, while in *Mtb*FtsEX, RipC can only bind from the side and is locked down by the lower lobe from the ECD hook facing the central axis. In summary, while in the pseudo-monas case, both ECD move in the same way in order to trap the EnvC partner in a symmetrical fashion; in mycobacterium the rearrangement of both ECD is asymmetrical, one of them remains in a similar disposition to that observed in the apo conformation (*Mtb*ECD 1 in Fig. 5c) while the other experiences a large conformational rearrangement. This movement is produced likely because of partial refolding of residues 49-55 from the large kink that, upon RipC recognition, folds as an α-helix enlarging TM1 and providing a straight conformation for the kink (*Mtb*ECD 2 in Fig. 5c) as detailed below. Severe mutation of this kink region residues showed obvious defects in RipC binding (Fig. 4e).

## Recognition of RipC through a "match and fit" mechanism

To better understand the mechanism by which RipC is recognized by FtsEX, we compared the structures of FtsEX in the presence or absence of RipC (Fig. 6a). The binding site for RipC is a large groove in the middle of the ECD domain that is formed when the two lower lobes are positioned on opposite sides away from the central axis of FtsEX. When RipC binds, there is an asymmetric response from ECD domains; one ECD domain remains stationary while the other rotates significantly to tightly lock RipC underneath (Fig. 6a). In particular, the ECD adjacent to α2 of RipC rotates about 54° with a shift of 26 Å observed from a tip residue Q112, positioning the Phe cluster (F61/F110/F113/F122) from the lower lobe perpendicularly on top of RipC to form a tight hook into a concave region of RipC (Fig. 6b). It is worth mentioning that the two specific features of the mycobacterium ECD (i.e. the disulfide bridge C73-C78 and the short α-helix, residues 64-68, both connecting the lower and the upper lobes) are essential in keeping the ECD hook conformation during rearrangement of the kink to lock RipC (Fig. 6c). In contrast, the other ECD domain adjacent to α1 shows no significant movement, but there are local conformational changes at the interface side-chains that lead to the ordering of the sidechains of the Phe cluster and their tight interaction with the α1 helix of RipC. After RipC recruitment, the two hook-like ECD domains become perpendicular to each other, along with the small loop and top region of TM helices, forming a joint cavity that locks RipC inside. In short, the mechanism of RipC binding involves a "match and fit" process, where one ECD matches and interacts with residues from α1, while the other ECD fits and rotates to lock α2 of RipC underneath.

## RipC activation is not triggered by ATP binding

The PG hydrolysis activity of the FtsEX-EnvC-AmiB system in *P. aeruginosa*[8], a gram-negative bacterium, has recently been established to be activated via ATP binding. As a comparative study, we tested if the *Mtb*FtsEX-RipC system could function through a similar mechanism by employing a PG hydrolysis assay with non-hydrolysable ATP analogues, such as ATP-γS (Fig. 7a). Contrary to the *Pae* system (Fig. 7a), we observed no significant PG hydrolysis activity in *Mtb* enzymes. This finding implies that *Mtb*FtsEX-RipC potentially employs a different mechanism from that of gram-negative bacteria in the

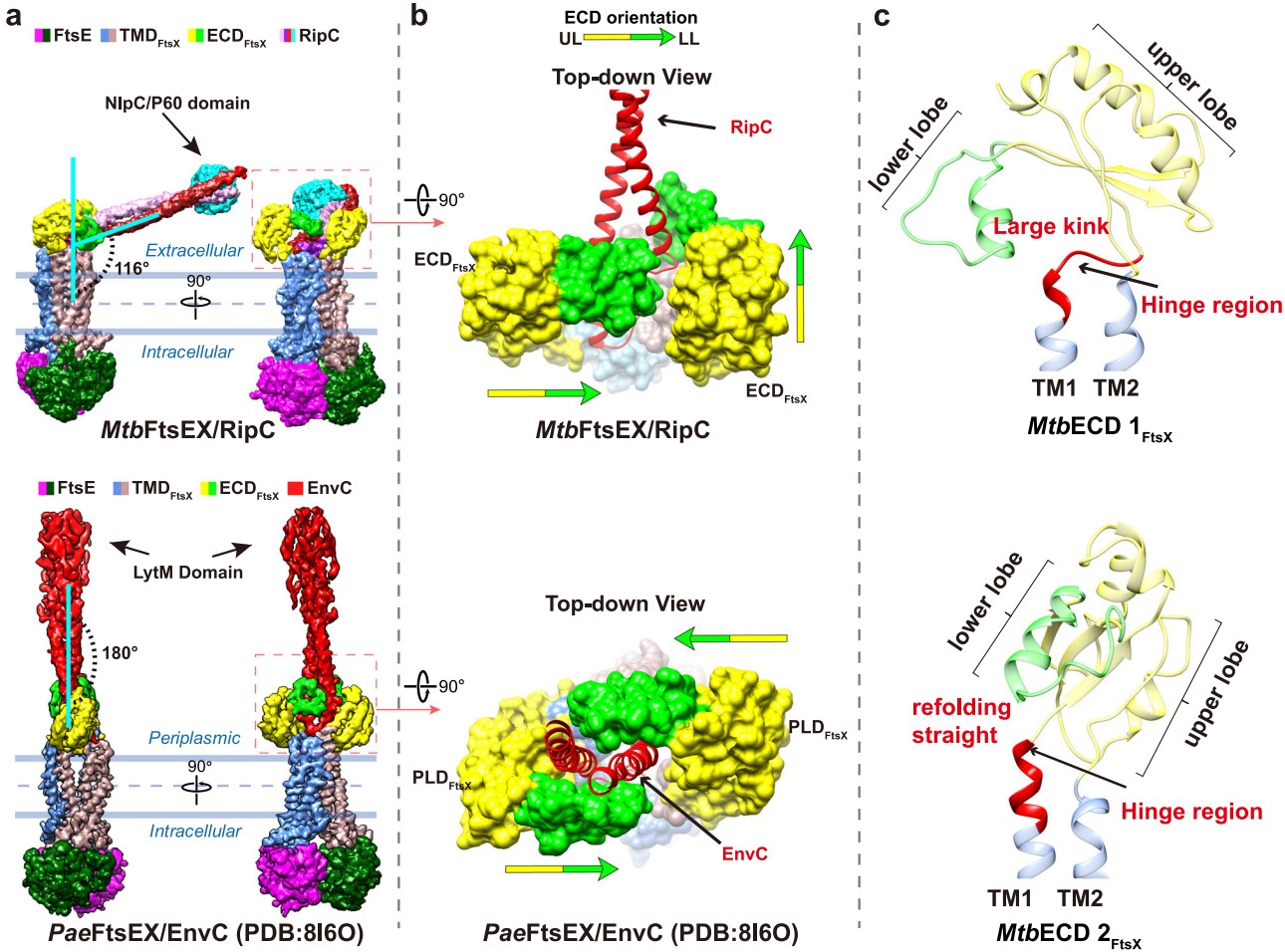

**Fig. 5 | A unique inclined binding mode of RipC by *Mtb*FtsEX. a** Comparison of the cryo-EM structure of RipC-bound *Mtb*FtsEX complex and the EnvC-bound *Pae*FtsX complex (PDB code: 8I6O) (EM density is shown in two orientations). EnvC binds close to the central axis of FtsEX and results in an elongated conformation, while RipC binds at a largely inclined angle with respect to the central axis of FtsEX. **b** Comparison of ECD/PLD orientation between RipC-bound *Mtb*FtsEX and EnvC-bound *Pae*FtsX-PLD (PDB code: 8I6O). The surface representations of the ECD/PLD domains are shown, with the upper lobe of the ECD/PLD colored yellow and the lower lobe colored green. RipC and EnvC are represented by ribbon and indicated by red color. UL upper lobe, LL lower lobe. The orientation of the ECD/PLD domain is indicated by arrow. **c** Comparison between the two ECD$_{FtsX}$ monomers from RipC-bound *Mtb*FtsEX complex. The two ECD monomers along with the connecting hinge region are represented by ribbon. the upper lobe of ECD$_{FtsX}$ is colored in yellow, and the lower lobe of ECD$_{FtsX}$ is in green, the TM1 and TM2 are in blue. Specifically, the hinge region from TM1 is highlighted in red, which exhibits highly dynamic features and determines the orientation of the connecting ECD.

activation of PG hydrolysis, where ATP hydrolysis may be crucial, not merely ATP binding.

To further investigate this hypothesis, we conducted a cryo-EM study on a *Mtb*FtsEX E165Q mutant, complexed with RipC, and in the presence of ATP (Supplementary Fig. 9). In this mutant, ATP can bind but not be hydrolyzed, thereby serving as a competent mimic of the ATP-bound state preceding ATP hydrolysis. Our cryo-EM analysis yielded a 3.9 Å map of the complex, wherein the EM density for both the bound ATP and the catalytic domain of RipC was well defined (Supplementary Fig. 10 and Fig. 7b). Detailed comparison of this structure with the ATP-free complex structure demonstrated that ATP binding brings the two FtsE subunits closer by about 3 Å (Fig. 7c, d), leading to a 2.7 Å restriction at the extracellular ECD domain (Fig. 7e). Regardless of these conformational differences, the tilting angle of RipC relative to the FtsEX central axis remained consistent, and no conformational changes were detected in RipC with the NlpC/P60 domain remain attached to the coiled-coil domain (Fig. 7f). This integrative functional and structural analysis suggests that in contrast to *P. aeruginosa*, ATP binding alone in *Mtb* is insufficient to trigger the activation of the hydrolase, indicating that further ATP hydrolysis may be required.

## Activation of FtsEX-RipC complex requires ATP hydrolysis

To study the role of ATP hydrolysis in the activation of RipC, we further conducted PG hydrolysis assays in the presence of multiple different ATP analogs (Fig. 8a). The results demonstrate that non-hydrolyzable ATP analogs such as ADP, ATPγS, and AMP-PNP do not affect FtsEX activity, confirming our previous findings (Fig. 7a). However, the complex becomes active in the presence of ATP-Mg$^{2+}$. Although the PG hydrolase activity was minimal, there was a noticeable increase in activity when both ATP and Mg$^{2+}$ were present, indicating the dependence of the complex's PG hydrolysis activity on ATP hydrolysis. Interestingly, when examining different ATP analogues, we observed that ADP-vanadate, which mimics a post-hydrolysis state of ATP also activates the PG hydrolysis activity of RipC. Consequently, these results suggest that the activation of PG hydrolysis seems to be through the post-hydrolysis state of FtsEX rather than the ATP-binding state.

We then used cryo-EM to study the FtsEX-RipC complex in the presence of 2 mM ATP-Mg$^{2+}$. Our results revealed two distinct conformations (Supplementary Fig. 11 and Supplementary Fig. 12a, b), both of which contained clear EM density for the bound nucleotides located at the NBD domain (Supplementary Fig. 11j). In the Type 1

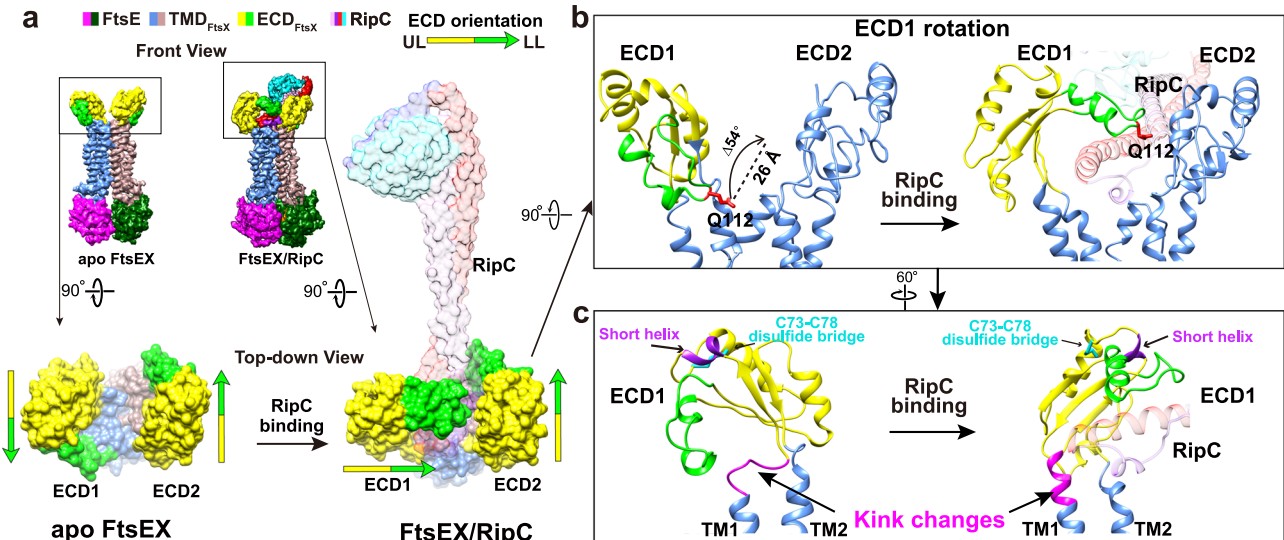

**Fig. 6 | Structural comparison of apo and RipC-bound *Mtb*FtsEX. a** Structural rearrangement of the ECD of FtsEX upon RipC binding. The full complex is in front view, the zoom-in views of ECD region is shown in top view and represented by surface. Color scheme: upper lobe in yellow, lower lobe in green; for RipC, α1 in plum, lip in purple, α2 in red, pro-rich linker in blue, the NlpC/P60 catalytic domain in cyan. UL upper lobe, LL lower lobe. **b** RipC binding triggers ECD1 to rotate in to grasp RipC, while ECD2 shows no significant movement. Structures are represented by ribbon, the Q112 is shown with sticks. RipC is with 70% transparency. **c** Switch between a straight and a kinked conformation of the hinge region of the TM1, controls the orientation of the linked ECD monomer for RipC binding. Color scheme: Kink region in magenta, short helix in purple, C73-C78 disulfide bridge in cyan. Bound RipC is with 70% transparency.

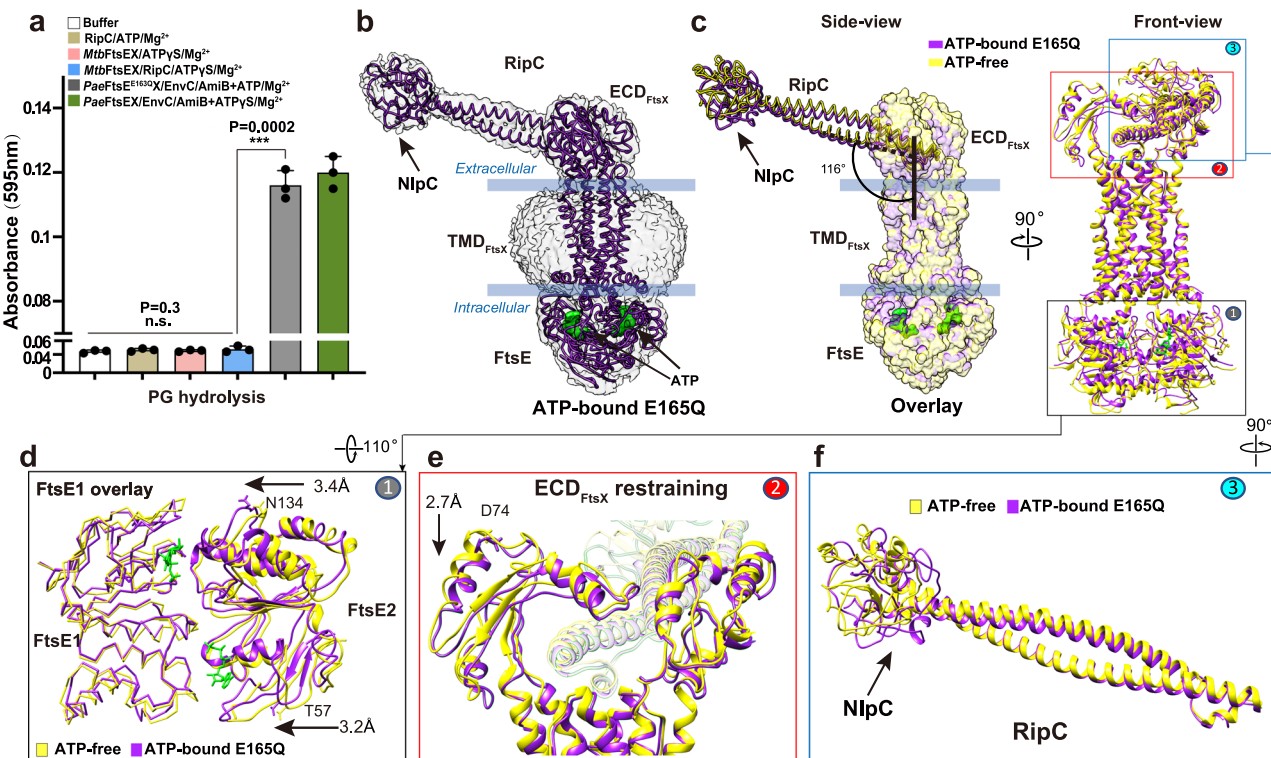

**Fig. 7 | Structural comparison of ATP-free FtsEX/RipC and ATP-bound FtsE^E165Q^X/RipC complexes. a** PG hydrolysis assay of FtsEX/RipC with non-hydrolysable ATPγS. Differs from *Pae*FtsEX system, no significant PG hydrolytic activity was observed for *Mtb*FtsEX in the presence of ATPγS. *n* = 3 replicates, error bars were presented as mean ± SD. Two-tailed unpaired *t*-test was applied; \*\*\**P* < 0.0005, n.s. no significant difference. The *P*-values are shown in figure and source data are provided as a Source Data file. **b** Cryo-EM density map of the ATP-bound FtsE^E165Q^X/RipC complex. Structures of FtsEX and RipC are represented by purple licorice. The EM density is in gray with 60% transparency, ATP are represented by green surface. **c** Superposition of the ATP-free FtsEX/RipC (in yellow) and ATP-bound FtsE^E165Q^X/RipC (in purple); Overall, no large conformational changes were observed. For Side view, the FtsEX components are shown as transparent surfaces, and the RipC hydrolase is represented in licorice form. For front view, FtsEX/RipC components are shown as ribbon. **d** Conformational changes of FtsE upon ATP binding. **e** Conformational changes in ECD observed during the transition from ATP-free to ATP-bound state. RipC is shown with 70% transparency. **f** RipC shows no significant movement upon ATP binding prior to ATP hydrolysis.

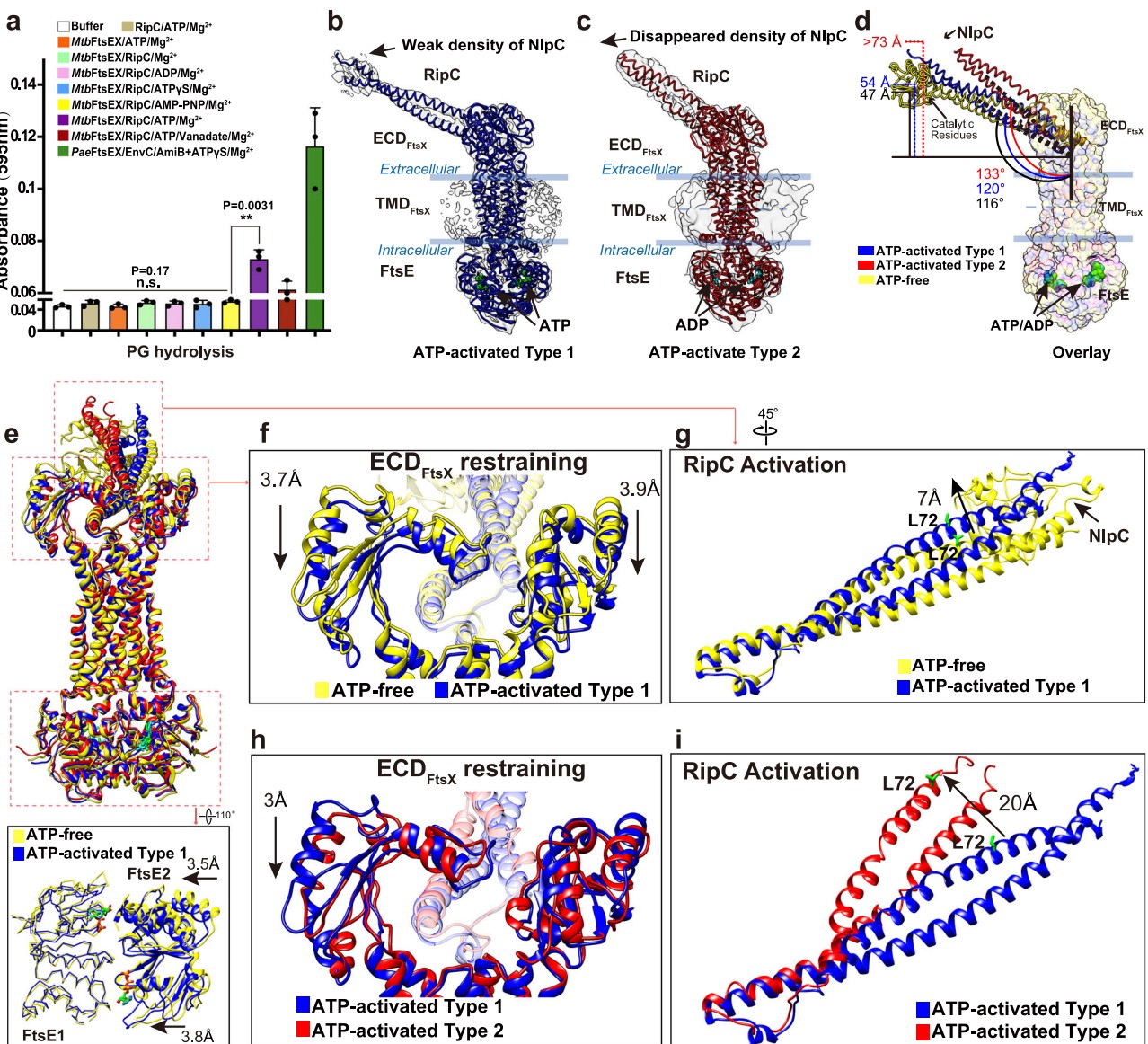

**Fig. 8 | Function-structure analysis of RipC activation. a** PG hydrolysis activity of RipC under different conditions. Buffer solution in the presence ATP-Mg$^{2+}$ was used as negative control and the *Pae*FtsEX/EnvC/AmiB as positive control. Only in the presence of ATP-Mg$^{2+}$ or trapped at a post-hydrolysis state by ATP-Mg$^{2+}$/vanadate, *Mtb*FtsEX/RipC showed minimal PG hydrolysis activity. $n = 3$ replicates, error bars were presented as mean ± SD. Two-tailed unpaired *t*-test was applied; **$P < 0.005$, n.s. no significant difference. The *P*-values are shown in figure and source data are provided as a Source Data file. **b, c** Cryo-EM density map of the ATP-activated RipC-bound *Mtb*FtsEX complex is shown in grey with 60% transparency surface with two conformations, Type 1 (**b**) and Type 2 (**c**). In the Type 1 state, there is a weak density of NlpC/P60 domain, while in the Type 2 state, the corresponding density completely disappears. The model for the Type 1 conformation is shown in blue, and the model for the Type 2 conformation is shown in the red. **d** Side-view of super-position of the RipC-bound *Mtb*FtsEX complex in three different states: ATP-free (in yellow), ATP-activated Type 1 (in blue), and ATP-activated Type 2 (in red); the tilt of RipC upon ATP hydrolysis is highlighted. The FtsEX components are shown as transparent surfaces, and the RipC hydrolase is represented in licorice. **e**, Upper: Front-view of the overlay of the RipC-bound *Mtb*FtsEX complex in three different states: ATP-free (in yellow), ATP-activated Type 1 (in blue), and ATP-activated Type 2 (in red); Lower: conformational changes of FtsE upon ATP binding. **f, g** Conformational changes in ECD (**f**) and RipC (**g**) observed during the transition from ATP-free state to ATP-activated Type 1 state. The ECD domains are shown in ribbon, while RipC is represented in transparent ribbon (70%). Structural differences are labeled, with the ATP-free state in yellow, and the ATP-activated Type 1 state in blue. **h, i** Conformational changes in ECD (**h**) and RipC (**i**) were observed during the transition from ATP-activated Type 1 state to ATP-activated Type 2 state. Structural differences are labelled, with the ATP-activated Type 1 state in blue, and the ATP-activated Type 2 state in red.

conformation, which was resolved at 4 Å with EM density compatible with a full ATP molecule, only partial density for the NlpC/P60 region was observed (Fig. 8b), which is much weaker compared to the RipC structure in its autoinhibited state (Supplementary Fig. 12c). Further flexibility analysis (Supplementary Movie 1) revealed that the particles within this class exhibited varying densities of the NlpC/P60 domain. Some particles displayed a higher density of the NlpC/P60 domain, while others had a significantly lower density. This finding confirms the presence of a highly flexible NlpC/P60 domain within the included

particles and indicates part of them started to unlock the NlpC/P60 domain for activation. These observations suggest that the catalytic domain is beginning to dissociate from the α1 of the CCD. In contrast, the NlpC/P60 domain was completely absent in the Type 2 conformation, resolved at 5.7 Å (Fig. 8c) and with EM density compatible with an ADP molecule (Supplementary Fig. 11j). Consistent with this, no density corresponding to the NlpC/P60 domain could be observed at the location associated with the autoinhibited state based on our flexibility analysis (Supplementary Movie 2), indicating that RipC was

activated and the NlpC/P60 domain was released. Although the identity of the bound nucleotide in the Type 2 conformation was difficult to distinguish at the current resolution, we propose that it represents the post-hydrolysis state with ATP hydrolyzed to ADP, as we observed large conformational changes. While the Type 1 conformation largely resembles that of the pre-hydrolysis state observed in the ATP-bound MtbFtsEX-RipC complex, despite the increased flexibility of the NlpC/P60 domain. We thus propose that the Type 1 conformation represents an intermediate state before RipC is fully activated upon ATP binding, while the Type 2 conformation represents a fully activated state with ATP hydrolyzed, suggested by the disappearance of the NlpC/P60 density.

While the overall structure of FtsEX in the presence of nucleotide showed no significant conformational changes compared to the RipC-bound structure in the absence of ATP, the release of NlpC/P60 and a clear tilting of the CC domain towards the PG side away from the membrane were observed (Fig. 8d). In the ATP-free state, the bound RipC was tilted at ~116°, with the NlpC/P60 domain being well-folded and grasped by the tweezers-like CCD domain and pro-rich linker that is specifically restrained by the N-terminal residues of α1. Upon ATP binding, the tilting angle of the Type 1 conformation increased to 120°, and in the Type 2 conformation, it increased further to 133° towards the PG layer, and with the NlpC/P60 domain released and no longer restrained by α1 (Fig. 8d). Consequently, assuming no further drastic conformational changes, the released NlpC/P60 domain would be positioned ~70–80 Å away in the Type 2 conformation (Fig. 8d).

### "Inside-out" mechanotransmission mechanism in the activation of RipC

Through superposition of the Type-1, Type-2 conformations and the ATP-free FtsEX-RipC structures, we identified a series of conformational changes that occur during RipC activation (Fig. 8e–i). First, upon ATP binding, the two FtsE subunits come into close proximity,

resulting in a shift of one NBD by about 3.5 Å (Fig. 8e), as observed in both Type 1 and Type 2 conformations. Engagement of the cytosolic NBD leads to minor conformational changes at the hinge region of the transmembrane helices (TM1 & TM2) that connect with the ECD. This causes both ECDs to squeeze towards bound RipC underneath for about 3.8 Å (Fig. 8f), resulting in a Type 1 conformation where the CC domain of RipC is raised by about 7 Å (Fig. 8g). An additional 2–3 Å of squeezing of both ECDs leads to the transition from Type 1 to Type 2 conformation (Fig. 8h), propagating the signal through the CC domain and eventually releasing the NlpC/P60 domain by tilting the CC domain of RipC for another 20 Å (Fig. 8i). While the flexibility of the released NlpC/P60 domain precludes direct visualization of the catalytic domain, combined with the uncoupling of the blocking helix of α1 from the catalytic site, indicates that RipC is likely activated for PG lysis.

## Discussion

Structural characterization of the FtsEX system alone and in complex with RipC, and in the presence or absence of ATP allowed us to present a model of how FtsEX controls the breakdown of PG at the division site in *M. tuberculosis* (Fig. 9). When ATP is absent, the FtsEX-RipC complex adopts a folded conformation with the NlpC/P60 domain inhibited. Interestingly, unlike the gram-negative systems identified thus far[8,30], the coiled-coil containing protein is recognized upon asymmetric rearrangement of the ECD of FtsX resulting in a unique inclined binding mode of RipC. Differences are also observed in the activation process between gram-negative systems and *M. tuberculosis*, mere ATP binding proves insufficient to trigger the activation of RipC. Only in the presence of ATP-Mg²⁺, FtsEX undergoes a series of conformational changes that are initiated upon ATP binding/hydrolysis and propagate through the transmembrane domain (TMD) of FtsX to its ECD (Supplementary Movie 3). These ECD changes result in a catalytic domain-unlocked conformation of RipC, which frees the NlpC/P60 domain and

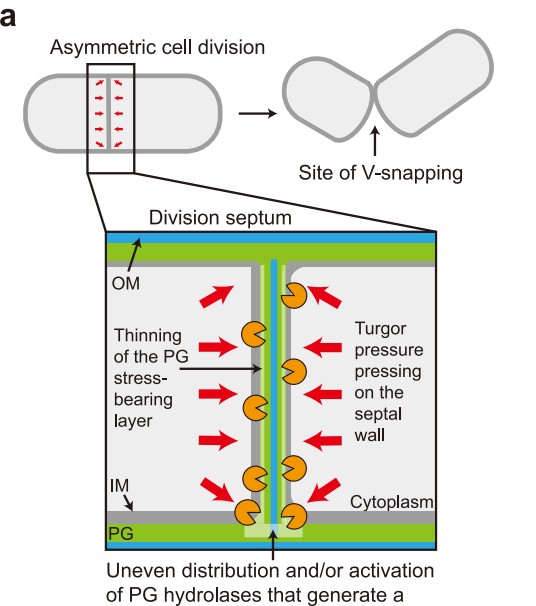

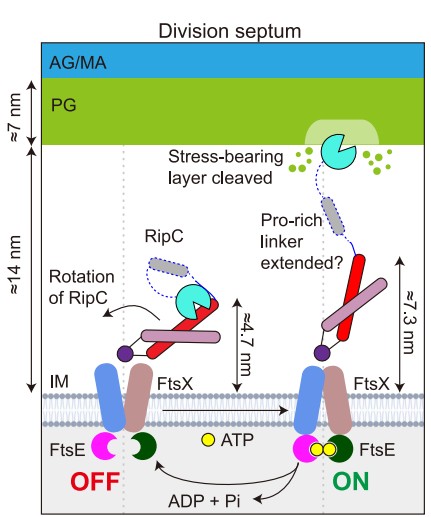

**Fig. 9 | FtsEX/RipC complex involved peptidoglycan hydrolysis in *Mycobacterium tuberculosis*. a** Model showing uneven distribution and/or activation of PG hydrolases that generate a breaking point for V-snapping in *Mycobacterium tuberculosis*. **b** Working model for FtsEX regulated RipC activation in *Mycobacterium tuberculosis*. This model illustrates the transition of the FtsEX/RipC complex from its autoinhibited state (left) to its activated state (right) during peptidoglycan hydrolysis at the division site. Initially, the complex is in its autoinhibited state and

unable to cleave the peptidoglycan layer. However, ATP binding and hydrolysis at the cytosolic side initiates a series of conformational changes that ultimately lead to the activation of the complex for peptidoglycan hydrolysis at the division site. Once the hydrolysis is complete, ATP hydrolysis recycles the complex back to its auto-inhibited state. Color scheme: FtsX is indicated in cornflower blue/rosy brown, FtsE is indicated in magenta/dark green. For RipC, α1 in plum, lip in purple, α2 in red, pro-rich linker in blue, the NlpC/P60 catalytic domain in cyan.

activates its catalytic site for PG cleavage. The release of hydrolyzed nucleotide then returns the FtsEX-RipC complex to its resting state, with the NlpC/P60 domain restrained by the N-terminal α1 helix. The precise roles of cofactors like SteA and SteB, as well as other divisome proteins, in modulating RipC activity are yet to be determined. Our structures reveal that an 11-aa long segment of the Pro-rich linker connecting the CCD with the hydrolytic domain is stabilized by a unique groove of the NlpC/P60 domain in the back of the active site, while the remaining part of this linker (39 aa) was disordered. The aa composition of the proline-rich linker and its partially disordered nature may have functional implications for the protein and its inter-actions with other cellular components. Proline-rich regions are known to preferentially adopt a specific helical conformation, an extended structure that facilitates transient intermolecular protein-protein interactions[36–38]. It is thus possible that the disordered Pro-rich linker serves as a hub for transient interactions with other divisome proteins to fine-tune cell wall degradation. In any case, our work reveals (i) that ATP binding at cytosol is the driving force to trigger cell-wall degra-dation in the extracellular space and (ii) how the mechanical force is transmitted from the FtsEX complex to the RipC to activate its catalytic domain.

In addition, we found that the ECD of the *Mtb*FtsEX complex is highly dynamic, which is derived from a hinge region located at the connecting part of the two adjacent TM1&TM2 (Supplementary Movie 4). This hinge region can adopt both a straight and a kinked conformation to orient the two ECD monomers. Additionally, the ECD in FtsEX can recognize two structurally distinct target sites with the same molecular surface. We suggest that flexibility in the lower lobe of *Mtb*FtsX is critical for the "dual-recognition", which is mediated by the "Four-Phe cluster". This finding is consistent with previous studies on the EnvC system in *E. coli*[30], which suggested that the aromatic residues Trp155 and Phe158 in FtsX are highly flexible and with studies on *S. pneumoniae* in which strains with mutations targeting the lower loop of the ECD FtsX (F126A, W123A) resulted in severe growth and mor-phology defects[10]. These aromatic residues provide plasticity by gen-erating a flat, planar, and hydrophobic interacting interface. Together, these structural features may contribute to the diversity of binding partners in the FtsEX systems.

Interestingly, while the ATP hydrolysis activity of *Pae*FtsEX was nearly undetectable until EnvC was bound[8], indicating its ATPase activation only occurs in the presence of this interaction, the *Mtb*FtsEX shows a substantial basal ATP hydrolysis activity, which was not reliant on its counterpart RipC. The high basal activity of *Mtb*FtsEX also does not appear to be crucial for its interaction with RipC. This finding reveals a clear difference in the regulation of PG hydrolysis between both systems and suggests that the ATPase activity may have other additional physiological functions in cell division as previously reported[39]. Otherwise, this activity could represent an unnecessary energy expenditure for Mycobacteria. Furthermore, the PG cleavage mechanism via the corresponding hydrolase differs between *Pae*FtsEX and *Mtb*FtsEX. In the case of *Pae*FtsEX, ATP binding alone is sufficient to activate PG cleavage by the FtsEX/EnvC/AmiB supercomplex[8]. However, *Mtb*FtsEX requires ATP hydrolysis, not merely binding, to activate RipC for PG cleavage. Related to this, a previous study indi-cates that the *Mtb*FtsX ECD can autonomously activate RipC PG hydrolase activity[15], possibly due to its flexibility and the presence of RipC in solution. This might precipitate an ECD-RipC complex forma-tion, bypassing the FtsEX complex and causing anomalous RipC acti-vation, similar to our recently reported *Pae*FtsEX study where EnvC alone triggers AmiB's PG hydrolase activity[8].

Despite the difference in hydrolase activation by ATP binding/ hydrolysis, the overall mechano-transmission mechanism appears to be conserved between various FtsEX systems. Yet, the distance between the PG hydrolase and the cytoplasmic membrane may vary. For example, the symmetrical arrangement of ECD *versus* the CC

domain of EnvC observed in *E. coli*[30] or in *P. aeruginosa*[8], together with the presence of an adaptor domain (LytM) and the associated amidase; warranties placement of the catalytic domain far into the periplasmic space, which may serve as a protective mechanism to avoid cleavage of the non-septal PG. By contrast, due to the inclination of the CC domain, RipC could only reach ~8 nm away from the cell membrane in its activated state. While considering the long Pro-rich linker, further extrusion of the catalytic domain could be expected and perhaps explain why the system can hydrolyze PG ~10 nm above the cell membrane. This inclination seems to be genuine for *M. tuberculosis* and may partly explain the "V-snapping" mode of cell division in *Cor-ynebacterineae*. By cleaving the stress-bearing layer of PG, FtsEX-RipC likely modifies the mechanical strength of the cell envelope such that tension is built through turgor pressure (Fig. 9). The subsequent uneven cleavage of PG results may ultimately result in a sudden energy release that snaps the mother cell into two daughters, similar to the asymmetric plant cell wall that promotes explosive seed dispersal[40]. The mechanism by which the septal cell wall is being cleaved unevenly, and how *Mtb*FtsEX avoids PG damages, remains unclear. It likely involves additional regulatory factors like SteAB. To this end, RipA tends to aggregate inside the cell and forms foci[14,16]. We speculate that RipC may also have this property as both proteins have similar domain organization[16]. Lastly, it remains unclear whether RipA, which also contains a CC domain, can interact with FtsEX despite the lack of interactions detected in a purified system[15]. Possibly, their interactions are bridged by SteAB. Their roles in controlling PG hydrolases in the context of FtsEX warrant future investigations.

## Methods

### Cloning and mutagenesis

The primers and templates utilized in this study are listed in Supple-mentary Table 1. To create the PCR amplicon containing ftsEX from *Mycobacterium tuberculosis*, Phusion DNA polymerase was used, fol-lowed by purification and digestion with NdeI and HindIII. After pur-ification through gel electrophoresis, the resulting product was ligated to pET28b and digested with the same enzymes to produce pET28b [PT7:his6-ftsEX]. The construction of pET28-sumo (PT7:his6-sumo-RipCΔ33) involved ligating pET28-sumo digested with BamHI and HindIII to a PCR amplicon synthesized by RipC F and RipC R, which were also digested with the same enzymes. Mutations to FtsE, FtsX, and RipC were generated utilizing the QuikChange kit (Agilent Tech-nologies), using pET28b and pET28-sumo as templates and the primers listed in Supplementary Table 1. The changes were then verified by DNA sequencing.

### Protein expression, purification of FtsEX complex and their mutants

FtsEX and FtsEX mutants were expressed in *E. coli* strain BL21(DE3) pLysS (Invitrogen) in lysogeny broth (LB) supplemented with 50 μg/ml kanamycin and 10 μg/ml chloramphenicol at 37 °C until the optical density at 600 nm (OD600) reached 0.8. After induction with 0.5 mM isopropyl-β-D-thiogalactoside (IPTG) at 20 °C for 16 h, cells were har-vested by centrifugation (Beckman) using a JLA9,1000 rotor at 5,180 xg, 20 °C for 15 min and stored at −80 °C.

To extract the membrane, the frozen cell pellets were thawed at 4 °C and resuspended in buffer A (50 mM Tris-HCl pH 8.0, 250 mM NaCl). The mixture was lysed using a high-pressure homogenizer (AH-100D, ATS), and the membrane was then solubilized using buffer B (50 mM Tris-HCl, 250 mM NaCl, 1.3% n-dodecyl β-D-maltoside (DDM), pH 8.0) for 2 h at 4 °C. After centrifugation at 100,000 xg for 30 min, the solubilized membrane fraction was incubated with TALON Super-Flow resin (Takara 635502), washed with buffer B supplemented with 20 mM imidazole, and 2 mM ATP/MgCl$_2$. The detergent was replaced by a peptide to produce peptidiscs using the "on-beads" method[41,42], the final peptide concentration used is at 1 mg/ml. FtsEX complex

reconstituted in peptidiscs was washed with buffer B containing 20 mM imidazole and eluted with buffer B supplemented with 200 mM imidazole. The purified protein was applied to a gel filtration chromatography column (Superose 6 increase 10/300) and eluted with buffer C (25 mM Tris-HCl, 150 mM NaCl, pH 8.0). Fractions containing the FtsEX complex were pooled, concentrated to 4 mg/ml, and stored at −80 °C.

## Protein expression, purification of RipC and its mutants
To express RipC and its variants, *E. coli* strain BL21(DE3) pLysS (Invitrogen) was used. The cells were grown in LB medium supplemented with 50 μg/ml ampicillin at 37 °C until the OD600 reached 0.6, and then induced by adding IPTG to a final concentration of 0.5 mM for 16 h at 20 °C. The cells were harvested by centrifugation (Beckman) using JLA9,1000 rotor at 5,180 xg, 20 °C for 15 min, and the pellets were stored at −80 °C. The frozen cell pellets were thawed at 4 °C in buffer A (50 mM Tris-HCl pH 8.0, 150 mM NaCl), and lysed using a high-pressure homogenizer (AH-100D, ATS). The mixture was centrifuged at 27,000 xg for 1 h at 4 °C. The supernatant was incubated with the TALON resin and washed with buffer A supplemented with 20 mM imidazole. After washing, the His-tagged SUMO protease Ulp1 was added to the resin-buffer mixture (0.1 mg Ulp1 per 1 L of culture) to cleave the SUMO tag at room temperature for 2 h. The cleaved samples were collected, concentrated, and applied to the Superose 6 increase 10/300 column equilibrated in buffer C. The fractions containing the target protein were identified by SDS-PAGE, pooled, concentrated to 3 mg/ml, and stored at −80 °C.

## Pull down assay
To investigate the interaction between FtsEX and RipC, we performed affinity chromatography using a TALON resin. Specifically, we applied 15 μg of FtsEX or its ATPase defective variants to the pre-equilibrated resin at room temperature for 15 min, using a Micro Gravity Column. After removing unbound FtsEX by washing the beads with 100 μl of buffer A containing 20 mM imidazole, we added 5 μg of RipC dissolved in 60 μl of the same buffer. We then washed the beads with buffer A supplemented with 20 mM imidazole in the presence or absence of ATP-Mg$^{2+}$. To elute the bound proteins and potential interacting partners, we used 20 μl of buffer A containing 300 mM imidazole and analyzed the eluted proteins by SDS-PAGE.

## Reconstitution of FtsEX-RipC complexes
To reconstitute the FtsEX-RipC complex, we added excess RipC (3-fold the amount of FtsEX) to the resin-bound FtsEX-peptidiscs during purification. The mixture was incubated at room temperature for 15 min, after which unbound EnvC was removed by washing with buffer A containing 20 mM imidazole. The FtsEX-RipC complex was eluted from the resin using buffer A with 200 mM imidazole. The eluted protein was then further purified by gel filtration (using a Superose 6 increase 10/300 column) in buffer B. Fractions containing the FtsEX-RipC complex were identified by SDS-PAGE, pooled, concentrated to 3 mg/ml, and stored at −80 °C.

## ATP hydrolysis assay
To measure the ATPase activity of FtsEX and its variants, we used the malachite green phosphate assay[43]. We added 2 μg of protein to a 20 μl reaction buffer containing 25 mM Tris-HCl pH 8.0, 150 mM NaCl, and 2 mM ATP/MgCl$_2$. To determine the Michaelis-Menten constants, we added different amounts of ATP/MgCl$_2$ (ranging from 0.05 to 4 mM) to initiate the reaction. After incubating the mixture at 37 °C for 10 min, we added 80 μl of Malachite Green-Ammonium Molybdate (MG-AG) solution, and immediately vortexed the mixture to inactivate the enzyme. After 2 min, we added 10 μl of a 34% (w/v) sodium citrate solution and vortexed the mixture again. We then measured the absorbance at 650 nm after 5 min. The amount of inorganic phosphate

(Pi) released was calculated using a standard curve generated by a known amount of Pi from a K$_2$HPO$_4$ solution. We fitted the data to the Michaelis-Menten equation using Microsoft Excel and GraphPad Prism 9.

## Isolation of peptidoglycan
Peptidoglycan isolation was performed as previously described[44,45], with minor modifications. Briefly, 400 ml of HMS0002 strain[46] was cultured in brain heart infusion broth at 37 °C in 5% CO$_2$ until OD$_{600}$ reached ~0.4. Cells were harvested by centrifugation at 5000 xg for 10 min at room temperature and resuspended in 15 ml of 1x PBS. The cell suspension was immediately transferred to 10 ml of boiling 10% SDS and boiled for 30 min, followed by centrifugation at 10,000 xg for 10 min at room temperature. The crude peptidoglycan pellets were washed 4 times with 0.1 x PBS and once with 2 ml of water. The pellets were then resuspended in 1 ml of reaction buffer (50 mM Tris-HCl, pH 7.4, 1 mM MgCl$_2$), and RNase I (10 mg/ml, EN0531) and DNase I (M0303S) were added (2 μl each). The mixture was incubated at 37 °C for 1 h, and the digested peptidoglycan was collected by centrifugation at 20,000 xg for 5 min at room temperature. The peptidoglycan was then resuspended in 1 ml of 1x PBS, and 200 μg of trypsin (Sigma) was added. The mixture was incubated overnight at 37 °C, and the peptidoglycan was collected by centrifugation. The pellets were washed three times with 1 ml of 1% SDS, once with 1 ml of 8 M LiCl with 15 min incubation at 37 °C, once with 1 ml of 0.1 M EDTA with 15 min incubation at 37 °C, and finally three times with 1 ml of water. The purified peptidoglycan was resuspended in 1 ml of water and stored at 4 °C.

## Dye release assay
To label PG with Ramazol brilliant blue, we followed the procedure outlined in[44]. Briefly, we mixed 478 μl of PG suspension with 56.3 μl of 0.2 M Ramazol Brilliant Blue R (Sigma R8001 in water), 28.2 μl of 5 M NaOH, and 456.3 μl of water. The mixture was then incubated at 37 °C for 18 h, and labeled PG was collected by centrifugation at 20,000 xg for 5 min at room temperature. After extensive washing with water until the supernatant became colorless, the labeled PG was resuspended in ddH$_2$O and stored at 4 °C.

To remove any unbound dye, the RBB-labeled PG was washed once with water before use. For PG digestion, 20 μl of RBB-labeled PG was mixed with the indicated proteins dissolved in 80 μl of reaction buffer (25 mM Tris-HCl pH 8.0, 150 mM NaCl, 2 mM ATP-Mg$^{2+}$) and incubated at 37 °C for 2 h. The final protein concentration was ~2 μM. The reaction was stopped by heating the mixture at 95 °C for 5 min, followed by centrifugation at 20,000 xg for 5 min at room temperature. The supernatant (80 μl) was then carefully transferred to a 96-well transparent plate, and the absorbance at 595 nm was measured.

## Electron microscopy sample preparation and data acquisition
To prepare the sample, 3.5 μl of the protein sample at a concentration of 3 mg/ml was applied to Quantifoil holey carbon grids (R1.2/1.3, 400 mesh) that were glow-discharged. To obtain the ATP-bound complex, the sample was incubated at 37 °C for 8 min with 2 mM ATP-Mg$^{2+}$ before freezing. The grids were blotted for 3.5–4 s with 100% relative humidity and then plunge-frozen in liquid ethane that was cooled by liquid nitrogen using a Vitrobot System (Gatan).

Cryo-electron microscopy data was collected at liquid nitrogen temperature using a Titan Krios electron microscope (Thermo Fisher Scientific), equipped with a K3 Summit direct electron detector (Gatan) and GIF Quantum energy filter. All cryo-EM movies were recorded in counting mode with SerialEM4[47] with a slit width of 20 eV from the energy filter. Movies were acquired at nominal magnifications of 81k x, which corresponded to a calibrated pixel size of 1.06 Å on the specimen level. The total exposure time for each movie was 6 s, resulting in a total dose range of 38-45 electrons per Å$^2$, fractionated

into 34 frames. Additional electron microscopy data collection parameters can be found in Supplementary Table 2.

## Electron microscope image processing

To process EM data, CryoSPARC[48] was used. Motion correction was performed using MotionCor2[49] on dose-fractionated movies collected with a K3 Summit direct electron detector. A sum of all frames from each movie was calculated using a dose-weighting scheme and utilized for all image-processing steps, except defocus determination, which was done using CTFFIND4[50] on the summed images from all movie frames without dose weighting. Particle picking was performed using the blob picker, followed by the template picker. For 2D and 3D classification and 3D refinement, the "2D classification", "Ab-initial Reconstruction", and "Heterogenous Refinement" methods were used, while "Homogeneous Refinement" and "Non-Uniform Refinement" were used for further refinement. The overall resolutions were estimated using the gold-standard criterion of Fourier shell correlation (FSC) = 0.143. The local resolution was estimated through "Local Resolution Estimation". 3D variability analysis was performed using CryoSPARC as well[51].

## Model building and refinement

Except for the Type 2 conformation in the presence of ATP-Mg$^{2+}$, all the maps in this study were resolved around 4 Å. These EM maps provided sufficient quality of density (<3.5 Å) for the eight TM helices (Supplementary Fig. 2 and Supplementary Fig. 5), thereby enabling direct de novo model building in Coot[52]. For the ECD, FtsE, and the RipC part, however, the density did not reach the quality observed in the TMD, with only some sidechains resolved. For these parts, the modeling process leveraged an initial model generated by AlphaFold[53]. This preliminary model was then rigid-body fitted into the cryo-EM map in UCSF Chimera[54] and subsequently manually adjusted within Coot[52].

Model building was facilitated through the application of standard covalent geometry restraints, "minimization_global," "local_grid_search," "adp," and "Ramachandran restraints" using "Real-space refinement" in Phenix. Each structure was finalized through multiple rounds of manual model building in Coot and refinements executed using "Real-space refinement" in Phenix[55].

For the 5.7 Å EM map corresponding to the Type 2 conformation in the presence of ATP-Mg$^{2+}$, no side-chain density was resolved. However, the secondary structures were sufficiently traceable, leading to the construction of a polyalanine model. This involved removing all sidechains from the ATP-bound complex structure obtained in this study, followed by a "real-space refinement" in Phenix[55]. The refinement incorporated "rigid_body" and "simulated_annealing" restraints, as well as "minimization_global," with "secondary structure restraints" information from the ATP-bound complex structure applied.

We generated restraints for ATP using the phenix_elbow program using its isomeric SMILES string files obtained from the PDB Chemical Component Dictionary through Ligand Expo. We manually docked the ligands into the cryo-EM maps in Coot, followed by iterative real-space refinements in Phenix. Final models were validated with statistics from Ramachandran plots, MolProbity scores, and Clash scores in Phenix (see Supplementary Table 2 for details). We used UCSF Chimera to generate figures.

## Statistics and reproducibility

No statistical method was used to predetermine sample size. For cryo-EM raw micrograph screening, we excluded images based on their quality and the presence of ice contamination. As for particle selection, we used criteria that depended on the quality of both the generated 2D class averages and the 3D map. The experiments were not randomized. The Investigators were not blinded to allocation during experiments and outcome assessment.

## Reporting summary

Further information on research design is available in the Nature Portfolio Reporting Summary linked to this article.

## Data availability

Five three-dimensional cryo-EM density maps of *Mtb*FtsEX and its complexes with RipC in the presence and absence of bound ATP have been deposited in the Electron Microscopy Data Bank under accession codes: EMDB-35362 (ATP-free FtsEX); EMDB-35363 (ATP-free FtsEX/RipC); EMDB-36304 (ATP-bound FtsE$^{E165Q}$X/RipC); EMDB-35364 (ATP-bound FtsEX/RipC complex Type 1); EMDB-35437 (ADP-bound FtsEX/RipC complex Type 2). Five atomic models have been deposited in the Protein Data Bank under accession codes 8IDB (ATP-free FtsEX); 8IDC (ATP-free FtsEX/RipC); 8JIA (ATP-bound FtsE$^{E165Q}$X/RipC); 8IDD (ATP-bound FtsEX/RipC complex Type 1); 8IGQ (ADP-bound FtsEX/RipC complex Type 2). Previously published structures used in this study for comparison were obtained from the Protein Data Bank under accession codes 8I6O (EnvC-bound *Pae*FtsEX); 5LIL (*Aa*MacB); 7ARK (*Ec*LolC); 4N8N (*Mtb*FtsX$^{ECD}$); 3NE0 (catalytic domain of RipA); 3PBI (catalytic domain of RipB); 8AUC (catalytic domain of Cg1735). The structures used for modelling were obtained from the Alphafold with codes: AF-A0A045JB98-F1 (*Mtb*FtsE); AF-A0A045GRS5-F1 (*Mtb*FtsX); AF-P9WHU3-F1 (*Mtb*RipC). The source data underlying Fig. 1a, Fig. 1c,d, Fig. 7a, Fig. 8a, Supplementary Fig. 1a, Supplementary Fig. 1f, Supplementary Fig. 4e, Supplementary Fig. 8, Supplementary Fig. 9e, and Supplementary Fig. 11f, g are provided as a Source Data file. Source data are provided with this paper.

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

## Acknowledgements

We are grateful to Thomas Bernhardt at Harvard Medical School, for helpful discussion and comment on the project. We are grateful for the kind assistance of Wan Zhen Chua from the Sham Lab and Atsushi Taguchi from Osaka University, Japan, for providing PG for our PG hydrolysis assay. We thank Zongli Li at Harvard Medical School and Yumei Wang at the Institute of Physics in China, for their help in initial sample screening. We also thank all Luo lab members for their constructive comments on this study. This work is supported by grants from the National University of Singapore Start-up grant (the Ministry of Education Academic Research (MOE) Tier 1 Grants 21-0053-A0001, 22-3449-A0001, and 22-3448-A0001 to M.L., NUHSRO/2017/070/SU/01 to L.-T.S.), and the National Research Foundation Fellowship (NRF) (NRFF11-2019-0005 to L.-T.S.), the MOE Tier 2 Fund (MOE-T2EP30222-0015 to M.L., MOE-T2EP30220-0012 to L.-T.S.). In Spain work was supported by grants PID2020-115331GB-100 funded by MCIN/AEI/10.13039/501100011033 and CRSII5_198737/1 (Swiss National Science Foundation) to J.A.H..

## Author contributions

J.L. and X.X. performed all experiments; J.S. helped with EM data collection; J.A.H., L.-T.S., and M.L. supervised the research and analyzed data, and M.L. wrote the paper with help from all authors.

## Competing interests

The authors declare no competing interests.
