## [Peer Review File · Nature Communications]

Regulation of the Cell Division Hydrolase RipC by the FtsEX system in *Mycobacterium tuberculosis*Reviewer #1 (Remarks to the Author):

Dear Li Jianwei and co-authors,

I have read your article entitled "The Mycobacterial FtsEX complex regulates the activation of peptidoglycan hydrolase RipC" and found it to be a valuable contribution to the field. The cryo-EM structures and careful analysis provided in your work are particularly noteworthy. In addition, I appreciate how in-vitro experiments on mutants and WT proteins, based on initial structures, led to other structures that provided valuable insights.

Your main claims are well supported by the structures, their analysis, and the validation studies using mutants and different in-vitro approaches. Furthermore, your work reveals an activation mechanism different from previously studied systems and could impact research towards new therapeutic strategies against *Mycobacterium tuberculosis*.

While your methodology is sound, I suggest providing more detailed information about your cryo-EM data processing procedures. For example, there is no information regarding CTF parameters estimation and refinement of these parameters during 3D refinement. Please include details of the most important (not all!) parameters used during the different stages of refinement so anyone can reproduce your work. It would also be advisable to include rough numbers of movies and particles at the beginning, during, and at the end of data processing.

Additionally, I noticed that all models have severe problems with clashes and geometry of proteins and ATP molecules. Although these were still non-final models, addressing these issues before the deposition of maps and models and acceptance of this work is essential.

Overall, your work is sound and provides novel structures and insights that will aid in understanding these complex biochemical cascades. Nevertheless, I have a few comments that should be addressed:

1. Abstract: *Mycobacterium tuberculosis* should have a lowercase "t."
2. Paragraph between lines 113-125: Last sentence ("These findings suggest that FtsEX forms a stable complex with its cognate hydrolase") should be moved up to avoid confusion. If I understood correctly, this statement relates to the initial part of the paragraph (until line 117). Decoupling it from the argumentation makes it confusing, in my opinion.
3. Although structures are informative and maps are good enough to build their corresponding models with enough confidence, $\sim 4\text{\AA}$ resolution is at the limit for correctly building side chains. Therefore, I believe the quality statements (e.g., lines 135-136, 185-186, etc.) should be toned down and acknowledge that the resolution has some limitations. Together with this, it would help to include in the model building methods section details about restraints, etc., during model building in the more complex areas.
4. Line 159-160: I do not understand the logic behind "the lower lobe of FtsEX is notably smaller in size, which suggests that it may have evolved to accommodate larger protein partners for binding." Are there any other pieces of information supporting this statement? Any correlation between the size of the lower lobe and the different sizes of binding partners?
5. Line 195: I don't think the analogy between the shape of the Ec and Mtb complexes with "I (capital i)" and " Γ (capital gamma?)" is the most straightforward one. I don't have a strong opinion about this, but it has stopped my reading.
6. Line 197: It would be good to mention the "interesting questions" rather than the open statement.
7. I would appreciate it if authors could comment on the possibility that the different ECD domains arrangement in the Ec structure could be due to the lack of the rest of the FtsEX proteins in those structures. Could it still be possible that the binding mode in Ec is similar to the one observed here

if observed in the presence of full-length proteins?

8. Line 244-245: The final statement here is confusing ("and it appears that activation is easy by simply moving a1 away..."). What does "easy" mean here? Does it imply that the mechanism is simple? Or that the inhibition is not strong? I would advise rephrasing this for improved clarity.

9. Fig3F: the change in orientation of the ECD and RipC makes it a bit difficult to understand initially. I would advise either to try to provide an orientation that is more similar to Fig3A or to indicate somehow that this is a top view of the complex (if I understood the figure correctly.).

10. Please provide more details regarding cryo-EM data processing. Right now, judging the image processing strategy is difficult as not enough details are given.

11. Table S2: Bond length and angle RMSDs for the FtsEX structure seem surprisingly low compared to the other structures, while the resolution of maps is the same. Please comment on this and the high clash scores.

12. Please review your models to improve clashes and geometry.

13. ATP/ADP geometry also has issues. Please review.

14. Map vs Model FSC plots should also be included so readers can evaluate the quality of the models built into the cryo-EM maps.

15. Are the two maps resolved in the FtsEX-RipC-ATP sample discrete states? It is unclear from the analysis whether this is the case or whether they are two representative maps of a continuous range of motion. Perhaps this part could benefit from a flexibility analysis (on Cryosparc, cryoDRGN, RELION-multibody, or else) to provide more insights on this.

Reviewer #2 (Remarks to the Author):

Review of the manuscript titled: "Regulation of the Cell division Hydrolase RipC by the FtsEX system in M Tuberculosis" by Li et al. for Nature Communications.

This manuscript is of great interest because it addresses the function of FtsEX, a widely conserved type VII ABC transporter which controls muralytic activity in bacteria. While in Gram Negative bacteria, FtsEX acts through a periplasmic mediator, in Gram positive the activation of the peptidoglycan (PG) hydrolytic enzymes is direct. Understanding the molecular details on how the activation of these lytic enzymes happens in bacteria would lead to a better understanding of the mechanism of action of the Type VII ABC transporters group which function as signal transducers instead of transporters and could also lead to the design of novel antibiotics.

In this study describe cryo-EM structures of the Mycobacterium tuberculosis FtsEX (MtbFtsEX) alone but also in complex with the PG-hydrolase RipC and with or without ATP. Analysis of these structures brings some detailed information on the signal transduction mechanism leading to RipC activation when the FtsEX is hydrolysing ATP. It reveals a novel asymmetrical arrangement of the extracellular domains of FtsX not observed in other FtsX structures and a unique binding mode for RipC.

Major findings:

MtbFtsEX has a high basal ATPase activity

ATP binding is not required for RipC binding

Unique Binding mode of RipC to MtbFtsEX

The binding of RipC causes an asymmetrical rearrangement of the 2 ECD domains, with one ECD refolding residues 49-55 into an alpha-helix and changing the conformation of a kink in the TM1, making it straighter.

First structure of RipC

ATP binding affect activation of the complex:

Small increase in PG hydrolysis activity of the complex in presence of ATP

Addition of ATP reveals 2 new distinct conformations of the FtsEX-RipC complex by cryo-EM.

The writing of the manuscript is clear and explanations are clear. The results presented here seem overall convincing (I am not a structure expert). I do wish a couple controls (as indicated below with the ATPase mutant of FtsE) would have been addressed to make the study more convincing to a non structural biologist.

Major comments/questions:

Paragraph 3.1: Do mutations in residues 49-55 of the TM1 in FtsX (the kink which changes into an Alpha helix upon binding to RipC) would prevent RipC binding?

This could be tested along the pulldown assays Fig 3E and would confirm the structure results of the FtsEX-RipC complex.

Paragraph 5.1: the cryo-EM Study of the FTsEX-RipC complex in presence of ATP reveals 2 distinct conformations but their resolution is lower than for the Apo structure and it appears that the identity of the nucleotide bound in the Type 2 is unknown. Would using in these experiments the variants of FtsE described Fig 1 and mutated for either for ATP binding (D164A) or ATP hydrolysis (E165Q), provide answer about the nucleotide bound status of the Type 2 structure (after ATP hydrolysis or not) and could actually "freeze" the structures in intermediates that could be resolved at a higher resolution than what are presented in the manuscript? This could lead to a better description of the different conformational changes in function of the ATP cycle of FtsEX.

About the Model Fig 5 and in general: If RipC does not regulate FtsEX ATPase and FtsEX has a high basal ATPase activity in vitro, does that mean that RipC is activated all the time? In vitro PG hydrolysis seems to be limited (even if increased when FTsEX can hydrolyse ATP as shown fig S16). Is the FtsEX ATPase activity affected when PG is present and RipC activated? Maybe while RipC is hydrolysing PG it is kept in the active form and FtsEX cannot exchange its nucleotide as well, leading to a reduced ATPase activity.

What, in the authors minds, regulates FtsEX activation of RipC in vivo to prevent lysis of the cells if the PG-hydrolase is activity high? This is not clear from the manuscript

Other points:

I will need more explanations to understand the statement from these authors that their results provide the first structural basis for the "V-snapping" cell division mechanism in Mycobacterium (lines 98-99 and 396-398). Exactly how the arrangement of RipC onto the FtsEX leads to an uneven cleavage of the PG layers is not clear to the reader here. It seems that this uneven activity of RipC due to its unique positioning onto FtsX would be "too localized" to cause the uneven cleavage of the PG layer to create a "snap mechanism". Such a "V-snapping mechanism" as I understand it, would probably be caused by an asymmetric distribution of FtsEX around the septal plan, causing an asymmetric PG hydrolysis at the septum.... Please elaborate on a detailed mechanism for the V snapping as you see it caused by the asymmetric position of RipC onto FtsEX.

Line169: I think the authors mean Fig 1F as there is no Fig 1G in the manuscript

Reviewer #3 (Remarks to the Author):

Li et al describe the full-length complex of the widely conserved divisome complex FtsEX from *Mycobacterium tuberculosis* alone and in complex with the PG hydrolase RipC. The solid structural work sheds light on the mechanistic understanding of this complex that still remains poorly understood and is thus of importance to advance our molecular understanding of the divisome.

Although the work is important and suitable for readers of *Nat Comm*, it will require extensive rewriting and restructuring of the results and figures to be compatible with a non-specialist audience. It is written essentially as a structure report that relies heavily on supplementary figures for understanding and lacks an appropriate discussion of mechanisms and implications. The structural results are greatly underexploited in terms of analysis and should be put into the bigger picture of PG regulation during cell division.

Below I will point out my concerns and suggestions:

Some general points:

- The authors should review the description of the results and corresponding Figures to make the text amenable to a non-structural biologist audience.
- Most of the text cannot be understood unless you work through the supplementary Figures (there are 19). Please try to coordinate main text and figures in a narrative that is comprehensible for the reader.
- Results should not be numbered and subdivided into small paragraphs, it makes the reading difficult.
- The colors of many of the figures are very poor in contrast and make it impossible to interpret them correctly, in particular the main figures. Please use contrasting colors and check that the message described in the text corresponds to what is represented in the Figure.
- The Figure legends hardly describe anything. They should include the full information required to understand the figure.
- Authors should analyse conservation patterns and discuss these, both with regard to the FtsEX complex that is present all over bacteria and the Rip family of hydrolases from *Corynebacterineae*, in particular the similarities and differences between RipA and RipC where now there are full length structures from each family available.

Specific points:

- Line 47: remove "for the first time". Although this is the first full complex of the system there have been other studies providing partial molecular insights into activation mechanisms (ie Mavrici et al)
- I am not sure it is necessary to make statements about specific design of novel antibiotics, as it is not clear to what extent RipC is essential for mycobacterial growth and division. In case the authors want to include this, they should discuss the literature on the conditional essentiality of RipC.

Introduction:

- Rv numbers for FtsEX and RipC should be mentioned in the introduction
- Line 65: "any known substrate" is this referring to the comparison with the ABC transporters? This sentence should be extended and clarified.
- Lines 69-72: this should be written more precisely, RipA does not interact with FtsEX but with SteB. However the second part of the sentence is correct in the sense that the auto-inhibitory CC

is part of the hydrolase itself. Please rephrase

- Paragraph between lines 78-86: this paragraph should be rewritten to reflect more correctly what is known (or not known) in Corynebacterineae. For instance not all species within this suborder have asymmetric cell division, proteins such as CwsA are not conserved in all. Also, I would not refer to "a few novel proteins" maybe species-specific proteins is more appropriate?

- The introduction could do with a more complete coverage of the mycobacterial PG remodelling literature.

- Line 92-93, please add a reference to this statement.

Results:

- The results need to be re-organised into a style compatible with an article of high impact. The numbering is not needed and the general organization needs to be revisited. For instance the first result is 1. Biochemical reconstitution and cryo-EM study of FtsEX followed by 2. Cryo-EM structure of FtsEX followed by 2.1 Overall structure of FtsEX etc. All this should be one Result section on the structural characterization of FtsEX, starting from the purification, activity then the structure. Also there is a lot of technical information that precedes the actual finding, referring to supplementary data. This could be partially put into Materials and Methods. It makes the paper difficult to read and the important messages are drowned in the technical detail without a logical flow (for example lines 133-140).

- Line 109: the D164A mutant is unable to bind ATP and the E165Q mutant is unable to hydrolyse ATP. Unless I missed this, I could not see any evidence on binding. Please clarify.

- The authors should provide binding affinities for the FtsEX-RipC interaction in the presence of the different nucleotides or analogues. The affinities may (should) be affected (in particular with regard to what is later presented for activation and conformational changes upon hydrolysis).

- Lines 120-121: add standard deviations to values mentioned.

- Lines 123-125: it would be nice to make a more detailed description and interpretation of the results here, or even better in the discussion, together with how ATP hydrolysis affects activation. These sentences are not clear.

- Result 2.2 on the unusual ECD conformation (lines 147-179): these important paragraphs describe conformational changes and structural comparisons that are difficult to understand without very clear and well described figures. ALL the figures referred to here are supplementary figures (S5-8) except for Figure 1G that does not exist! The authors need to choose the correct figures to properly illustrate their message in the main Figures and use the supplementary for additional info that is not essential for understanding the main message. For instance, Figure S5 needs to be a main figure as it actually shows the structural detail in a comprehensive manner (except for the colors that are not contrasted enough, and the legend that needs to be improved).

- Figure 1 B: the gels seem to have been cut and put together, they should be separated by a blank space. Figure 1E: in the schematic all the abbreviations need to be explained in the legend; the colors used should be the same between the schematic and the structure model and be contrasted enough to be distinguished. Figure 1F, see comment below for Figure S7.

- Figure S7 needs to be represented differently to understand what the authors want to show (maybe cartoon with surface superposed (transparent) and side plus top view?)

- Results section 3. Same comments as above on reorganizing the flow and moving some of the technical details into the materials. Section 3.1: again, the authors start straight with a comparison when the result is the structure of the complex. This needs to be described first and then use this result to compare with what has been shown for EnvC. The first panel of the Figure 2A describing this section cannot be a structure that has previously been solved by another lab...

- In Figure 2, the active site of the catalytic domain is not well represented and hardly visible, the same goes for the details shown in Figures 2C-F. Please make these representations easier to understand.

- Line 200-201, this comparison is also done just above, please regroup all the comparison linked to this, and again this is a full paragraph of structural comparisons with only supplementary figures. Include the main comparisons in a comprehensive manner in the main paper.

- Lines 228-229: "typically": what does this mean? This should be explained more precisely and with the correct literature cited. If the authors refer to other RIP structures from Mtb, this should be described and compared in a figure. It is impossible to know for people who are not familiar with the details of these proteins what is being referred to here.

- The interaction of the catalytic domain with the coiled-coil and alpha1 should be compared (supplementary figure) with the closely related RipA structure from *C. glutamicum* (Gaday et al) that shows an autoinhibitory mechanism through the coiled-coil. This has important implications for conserved regulatory mechanisms in this important family of PG hydrolases of *Corynebacterineae*

- Lines 244-245: "the activation is easy by simply moving alpha1 away from NlpC/P60 domain". This is not a very scientific description, please rephrase.

- Figure 3E: Why is there less FtsX for some of the lanes? This needs to be normalized and quantified, because if there is less FtsX, there will also be less RipC...

- Figures 3A-D, again are very difficult to "read", the changing orientations are confusing, as well as the approximate color correspondence between the main model and the insets. Again, the lack of contrast in the colors makes it all difficult to interpret.

- Line 270 and Supplementary Figure S15: "...mutating these residues had no discernable effect on ATP hydrolysis". Are the authors referring to all the mutants shown in Figure S15? Indeed, in this figure the ATPase activities of the double mutant F110A/F113A and the triple mutant F61A/F110A/F113A look rather different. If these differences are not statistically relevant, this should be shown. Also, a negative control is missing in this figure.

- Result section 4.2 Figure 3F and G: these figures are difficult to interpret, they should be redrawn so that readers can easily understand the message the authors want to convey. It is unclear for example whether the two structures in 3F are shown in the same orientation. In Fig 3G there are too many things in the left panel, which might be separated, showing for instance the coil in exactly the same position and the tow domains in their apo or bound conformation (but not superposed). The right panel is completely unclear.

- Lines 286-289: What does this mean? Where does this come from? And in particular line 289: this sentence is incomprehensible.

- Section 5/5.1: PG activity test. The info in Figure S16 is essential information and needs to be included in the main text. It also lacks controls (buffer alone, positive control, FtsEX alone, FtsEX+RipC without vanadate). In particular, the FtsEX+RipC control is essential to support the author's claim that the nucleotide-free state represents the auto-inhibited form of FtsEX-RipC. The legend should be extended to include minimal information to interpret the figure (such as ATP concentration used, reaction conditions, etc).

- Mavrici et al (2014) showed that the ftsX ECD alone is capable of activating RipC PG hydrolase activity. The authors should discuss this observation in view of their findings that ATP hydrolysis is required to activate the complex

- Figure S18 is incomprehensible, I don't see how the authors can claim conformational rearrangements with these figures, please redo. Moreover, the supposed conformational

rearrangement may be an overinterpretation, as the observed differences might also be due to a lower occupancy compatible with partial detachment of the catalytic domain.

- Figure 4D, ininterpretable, again, due to colour choices, too small, too crowded.

- Results 5.2, again the figures are used inappropriately and are not clear or do not reflect the main text.

- Line 358: there is no evidence that allows to talk about unfolded RipC.

- Line 361: what are MctA and MctB? There is no reference or explanation.

- The discussion lacks precision and references. The authors should include a discussion on the Rip family in *Corynebacterineae*, analyse the similarities and differences, mention the fact that in this suborder of actinobacteria the most important hydrolase for cell separation is the SteA/B-controlled hydrolase RipA and not the FtsEX-controlled RipC, etc

A point-to-point reply addressing reviewers' comments: the responses are labeled in blue for clarity, and line numbers are provided based on the marked file.

REVIEWER COMMENTS

Reviewer #1 (Remarks to the Author):

Dear Li Jianwei and co-authors,

I have read your article entitled "The Mycobacterial FtsEX complex regulates the activation of peptidoglycan hydrolase RipC" and found it to be a valuable contribution to the field. The cryo-EM structures and careful analysis provided in your work are particularly noteworthy. In addition, I appreciate how in-vitro experiments on mutants and WT proteins, based on initial structures, led to other structures that provided valuable insights.

Your main claims are well supported by the structures, their analysis, and the validation studies using mutants and different in-vitro approaches. Furthermore, your work reveals an activation mechanism different from previously studied systems and could impact research towards new therapeutic strategies against Mycobacterium tuberculosis.

While your methodology is sound, I suggest providing more detailed information about your cryo-EM data processing procedures. For example, there is no information regarding CTF parameters estimation and refinement of these parameters during 3D refinement. Please include details of the most important (not all!) parameters used during the different stages of refinement so anyone can reproduce your work. It would also be advisable to include rough numbers of movies and particles at the beginning, during, and at the end of data processing.

Additionally, I noticed that all models have severe problems with clashes and geometry of proteins and ATP molecules. Although these were still non-final models, addressing these issues before the deposition of maps and models and acceptance of this work is essential.

Overall, your work is sound and provides novel structures and insights that will aid in understanding these complex biochemical cascades. Nevertheless, I have a few comments that should be addressed:

We appreciate your positive feedback regarding our work, particularly regarding the need for more detailed information on our cryo-EM data process, as well as the suggestions to enhance our models. We completely agree with your comments.

In response, our revised manuscript now presents a more thorough explanation of the cryo-EM data process as described in the EM workflow (**Fig. S1, S4, S9, and S11**), including critical steps such as motion correction, CTF calculation, particle picking, and 3D reconstruction. These additions ensure our method is transparent and reproducible.

Furthermore, we have conscientiously addressed all identified issues concerning clash and geometry in our models. These improved models have been validated by multiple tools such as Molprobity, etc.

1. Abstract: *Mycobacterium tuberculosis* should have a lowercase "t."

Corrected.

2. Paragraph between lines 113-125: Last sentence ("These findings suggest that FtsEX forms a stable complex with its cognate hydrolase") should be moved up to avoid confusion. If I understood correctly, this statement relates to the initial part of the paragraph (until line 117). Decoupling it from the argumentation makes it confusing, in my opinion.

Thank you for your feedback. We agree and have addressed your suggestion by moving and rephrasing the mentioned sentence at **L177** (L117 in the previous version).

3. Although structures are informative and maps are good enough to build their corresponding models with enough confidence, ~4Å resolution is at the limit for correctly building side chains. Therefore, I believe the quality statements (e.g., lines 135-136, 185-186, etc.) should be toned down and acknowledge that the resolution has some limitations. Together with this, it would help to include in the model building methods section details about restraints, etc., during model building in the more complex areas.

Thank you for your feedback, and we acknowledge your concerns. In response, we have made significant improvements. Firstly, we have refined the quality statements by clearly indicating the EM density quality across different sections of the map. We have also incorporated the relevant model building details in this section to provide comprehensive information. Furthermore, as suggested by Reviewer 3, we have now relocated this information to the Method and Material section at **L1247-L1266**.

4. Line 159-160: I do not understand the logic behind "the lower lobe of FtsEX is notably smaller in size, which suggests that it may have evolved to accommodate larger protein partners for binding." Are there any other pieces of information supporting this statement? Any correlation between the size of the lower lobe and the different sizes of binding partners?

Thank you for pointing this out. We fully agree that there is no necessary rationale behind these phenomena. Thus, we have taken this speculative statement out in the revised version.

5. Line 195: I don't think the analogy between the shape of the Ec and Mtb complexes with "I (capital i)" and "Γ (capital gamma?)" is the most straightforward one. I don't have a strong opinion about this, but it has stopped my reading.

Fully agreed. In the revised version, we have removed the mentioned analogy. Instead, we have used the terms 'elongated-' and 'inclined-' binding to describe the different binding modes in **L723-L726**, and specified the binding angle in **Fig. 5A**.

6. Line 197: It would be good to mention the "interesting questions" rather than the open statement.

Thank you for the suggestion, we have rephrased it in **L726-L728**.

7. I would appreciate it if authors could comment on the possibility that the different ECD domains arrangement in the Ec structure could be due to the lack of the rest of the FtsEX proteins in those structures. Could it still be possible that the binding mode in Ec is similar to the one observed here if observed in the presence of full-length proteins?

Excellent point. In response, we have revised our work between **L722-L775** by comparing the *Mtb*FtsEX structures to our recently published full-length FtsEX structures in *Pseudomonas aeruginosa* (doi: 10.1073/pnas.2301897120, *PNAS*), a gram-negative bacterium.

8. Line 244-245: The final statement here is confusing ("and it appears that activation is easy by simply moving a1 away..."). What does "easy" mean here? Does it imply that the mechanism is simple? Or that the inhibition is not strong? I would advise rephrasing this for improved clarity.

Thank you for pointing this out. We have rephrased this sentence in the revised version at **L688-L690**.

9. Fig3F: the change in orientation of the ECD and RipC makes it a bit difficult to understand initially. I would advise either to try to provide an orientation that is more similar to Fig3A or to indicate somehow that this is a top view of the complex (if I understood the figure correctly.).

Thank you for your suggestion. We have implemented the recommended changes to the previous **Fig. 3F**, which is now represented as **Fig. 6A**. The figure now includes a clear indication that it represents a top view. In addition, we have added an arrow to provide further orientation clarification in **Fig. 2, 5 & 6**.

10. Please provide more details regarding cryo-EM data processing. Right now, judging the image processing strategy is difficult as not enough details are given.

Thank you for the suggestion. Our revised manuscript now presents a more thorough explanation of the cryo-EM data process as described in the EM workflow (**Fig. S1, S4, S9, and S11**),

11. Table S2: Bond length and angle RMSDs for the FtsEX structure seem surprisingly low compared to the other structures, while the resolution of maps is the same. Please comment on this and the high clash scores.

Thank you for bringing this to our attention. It appears that during the later rounds of "Real-space refinement" in Phenix, "reference restraints" were inadvertently included, leading to the refinement of covalent geometry towards the supplied reference structure.

In our modified refinement, we have removed this restraint and have successfully addressed all the identified issues related to clash and geometry in our new models. These new models have been validated using various tools, including Molprobity. Furthermore, we have incorporated a detailed modeling strategy into the "Method and Material" section at **L1247-L1266** for clarification.

12. Please review your models to improve clashes and geometry.

We have optimized the refinement strategy to facilitate the model building process. After that, significant improvements have been made to reduce clashes and enhance the overall geometry of the models. Please check the attached validation reports or **Table 2** along with our revision submission.

13. ATP/ADP geometry also has issues. Please review.

Corrected, thanks.

14. Map vs Model FSC plots should also be included so readers can evaluate the quality of the models built into the cryo-EM maps.

Map vs Model FSC have been provided in all FSC plots in the revised version, please see **Fig. S1, S4, S9, and S11**. Thanks!

15. Are the two maps resolved in the FtsEX-RipC-ATP sample discrete states? It is unclear from the analysis whether this is the case or whether they are two representative maps of a continuous range of motion. Perhaps this part could benefit from a flexibility analysis (on Cryosparc, cryoDRGN, RELION-multibody, or else) to provide more insights on this.

The two maps resolved in the FtsEX-RipC-ATP sample should be two distinct states, as evidenced by the presence and absence of NlpC/P60 domain density from RipC by subsequent 2D or 3D classification. As suggested, further flexibility analysis using Cryosparc was conducted and confirmed this as well, the results are now included in **L863-L867, L871-L873, Movie S1 and Movie S2**.

To further prove this point and investigate the functional states of these two conformations, we have resolved an additional structure in the revised manuscript (**L808-L820, Fig 7**). This structure involves the MtbFtsEX E165Q mutant complexed

with both RipC and bound ATP. This mutant is capable of binding ATP but unable to hydrolyze it, thus capturing the complex in a pre-ATP hydrolysis state.

In this new structure, the NlpC/P60 domain of RipC exhibits well-folded density, constrained by the N-terminal helix. This indicates that ATP binding alone is insufficient to activate RipC. This structure closely resembles the Type 1 conformation, with minor conformational differences observed. Conversely, in the Type 2 conformation, the NlpC/P60 domain density of RipC is completely absent, accompanied by a large tilt in the CC domain. This suggests a post-hydrolysis state where RipC is activated.

Overall, our findings support the notion that the Type 1 conformation precedes RipC activation, while the Type 2 conformation represents a post-hydrolysis state with activated RipC.

Reviewer #2 (Remarks to the Author):

Review of the manuscript titled: "Regulation of the Cell division Hydrolase RipC by the FtsEX system in M Tuberculosis" by Li et al. for Nature Communications.

This manuscript is of great interest because it addresses the function of FtsEX, a widely conserved type VII ABC transporter which controls muralytic activity in bacteria. While in Gram Negative bacteria, FtsEX acts through a periplasmic mediator, in Gram positive the activation of the peptidoglycan (PG) hydrolytic enzymes is direct. Understanding the molecular details on how the activation of these lytic enzymes happens in bacteria would lead to a better understanding of the mechanism of action of the Type VII ABC transporters group which function as signal transducers instead of transporters and could also lead to the design of novel antibiotics.

We thank Reviewer #2 for laudatory comments and the excellent summary of the impact of this work.

In this study describe cryo-EM structures of the Mycobacterium tuberculosis FtsEX (MtbFtsEX) alone but also in complex with the PG-hydrolase RipC and with or without ATP. Analysis of these structures brings some detailed information on the signal transduction mechanism leading to RipC activation when the FtsEX is hydrolysing ATP. It reveals a novel asymmetrical arrangement of the extracellular domains of FtsX not observed in other FtsX structures and a unique binding mode for RipC.

Major findings:

MtbFtsEX has a high basal ATPase activity

ATP binding is not required for RipC binding

Unique Binding mode of RipC to MtbFtsEX

The binding of RipC causes an asymmetrical rearrangement of the 2 ECD domains, with one ECD refolding residues 49-55 into an alpha-helix and changing the conformation of a kink in the TM1, making it straighter.

First structure of RipC

ATP binding affect activation of the complex:

Small increase in PG hydrolysis activity of the complex in presence of ATP

Addition of ATP reveals 2 new distinct conformations of the FtsEX-RipC complex by cryo-EM.

The writing of the manuscript is clear and explanations are clear. The results presented here seem overall convincing (I am not a structure expert). I do wish a couple controls (as indicated below with the ATPase mutant of FtsE) would have been addressed to make the study more convincing to a non structural biologist.

We are sincerely grateful for your positive and encouraging feedback on our study. The time and effort you dedicated to reviewing our paper, as well as your recognition of its novel findings, is highly appreciated. In response to your request for control experiments, we have carefully conducted them and incorporated detailed results in the revised version of our manuscript. This has significantly improved the quality of our study. We would like to express our sincere thanks for your invaluable input.

Major comments/questions:

Paragraph 3.1: Do mutations in residues 49-55 of the TM1 in FtsX (the kink which changes into an Alpha helix upon binding to RipC) would prevent RipC binding? This could be tested along the pulldown assays Fig 3E and would confirm the structure results of the FtsEX-RipC complex.

As suggested, we engineered three mutants (I51A/Y52A/L53A, D54A/R55A, R49A/I51A/Y52A/L53A/D54A/R55A) in the Kink region. Notability, the affinity of I52A/Y53A/L54A and R49A/I51A/Y52A/L53A/D54A/R55A mutants to RipC is significantly reduced. Please check updated **Fig. 4E** and text in **L774-L775**.

Paragraph 5.1: the cryo-EM Study of the FtsEX-RipC complex in presence of ATP reveals 2 distinct conformations but their resolution is lower than for the Apo structure and it appears that the identity of the nucleotide bound in the Type 2 is unknown. Would using in these experiments the variants of FtsE described Fig 1 and mutated for either for ATP binding (D164A) or ATP hydrolysis (E165Q), provide answer about the nucleotide bound status of the Type 2 structure (after ATP hydrolysis or not) and could actually "freeze" the structures in intermediates that could be resolved at a higher resolution than what are presented in the manuscript? This could lead to a better description of the different conformational changes in function of the ATP cycle of FtsEX.

Thank you for your insightful suggestion. In the revised manuscript, we have included a new structure involving the MtbFtsEX E165Q mutant complexed with RipC and ATP. This mutant, capable of binding ATP but not hydrolyzing it, represents a pre-ATP hydrolysis state. Detailed results can be found at **L798-L820, Fig 7**.

In short, in this new structure, the NlpC/P60 domain of RipC exhibits well-folded density, indicating that ATP binding alone is insufficient for RipC activation. This closely resembles the Type 1 conformation, with minor conformational differences observed. The Type 1 conformation represents the pre-hydrolysis state, where ATP is trapped.

Conversely, the Type 2 conformation lacks NlpC/P60 domain density and exhibits a large tilt in the CC domain. This suggests that the Type 2 conformation represents a post-hydrolysis state, where RipC is activated and ATP is hydrolyzed to ADP.

Overall, the resolution of this new structure significantly strengthens and clarifies our findings. It supports the notion that the Type 1 conformation precedes RipC activation, while the Type 2 conformation represents a post-hydrolysis state with activated RipC. Additionally, this suggests a distinct mechanism from the FtsEX system in *Pseudomonas aeruginosa* in which ATP binding triggers amidase activation (doi: 10.1073/pnas.2301897120, *PNAS*). In *Mycobacterium*, ATP binding alone is insufficient for RipC activation, but ATP hydrolysis plays a critical role.

About the Model Fig 5 and in general: If RipC does not regulate FtsEX ATPase and FtsEX has a high basal ATPase activity in vitro, does that mean that RipC is activated all the time? In vitro PG hydrolysis seems to be limited (even if increased when FtsEX can hydrolyse ATP as shown fig S16). Is the FtsEX ATPase activity affected when PG is present and RipC activated? Maybe while RipC is hydrolysing PG it is kept in the active form and FtsEX cannot exchange its nucleotide as well, leading to a reduced ATPase activity. What, in the authors minds, regulates FtsEX activation of RipC in vivo to prevent lysis of the cells if the PG-hydrolase is activity high? This is not clear from the manuscript

Recently, there is new evidence suggesting the activity of the FtsEX system is likely controlled by SteAB (Rv1697 and Rv1698). These regulatory factors are likely involved in regulating the ATPase activity and/or the PG hydrolase activity in Mtb as well. This will be the subject of future studies. The complex also likely contains other components such as RipA. Our biochemical data present in this paper serve as an important first step in characterizing the system. The results show no evidence that RipC activates the ATPase activity of FtsEX or direct activation of the RipC hydrolytic activity by FtsEX and ATP, consistent with the involvement of additional regulatory factors that are required for PG hydrolysis.

Other points:

I will need more explanations to understand the statement from these authors that their results provide the first structural basis for the “V-snapping” cell division mechanism in *Mycobacterium* (lines 98-99 and 396-398). Exactly how the

arrangement of RipC onto the FtsEX leads to an uneven cleavage of the PG layers is not clear to the reader here. It seems that this uneven activity of RipC due to its unique positioning onto FtsX would be “too localized” to cause the uneven cleavage of the PG layer to create a “snap mechanism”. Such a “V-snapping mechanism” as I understand it, would probably be caused by an asymmetric distribution of FtsEX around the septal plan, causing an asymmetric PG hydrolysis at the septum.... Please elaborate on a detailed mechanism for the V snapping as you see it caused by the asymmetric position of RipC onto FtsEX.

The statement that our study provides the first structural basis for the V-snapping may be overreaching and thus we reworded it. Whether FtsEX and RipC are distributed unevenly in the Mtb cell is still an open question. The structure of FtsEX-RipC, unlike in *Pseudomonas aeruginosa*, is tilted by 64 degrees from the central axis (**Fig. 5A**). The tilting is partially relieved by ATP binding to 47 degrees, but it only reaches 7.3 Å (**Fig. 8D**). We hypothesize that this arrangement may lead to the cleavage of the stress-bearing layer of peptidoglycan, somehow leading to a turgor pressure-driven snapping of the daughter cells. We elaborated on how this could happen and modified Fig. 5 (see new **Fig. 9** in the revised version), to illustrate our speculation.

Line169: I think the authors mean Fig 1F as there is no Fig 1G in the manuscript

Corrected, thanks!

Reviewer #3 (Remarks to the Author):

Li et al describe the full-length complex of the widely conserved divisome complex FtsEX from Mycobacterium tuberculosis alone and in complex with the PG hydrolase RipC. The solid structural work sheds light on the mechanistic understanding of this complex that still remains poorly understood and is thus of importance to advance our molecular understanding of the divisome.

We thank Reviewer #3 for laudatory comments and her/his vision on the impact of this work.

Although the work is important and suitable for readers of Nat Comm, it will require extensive rewriting and restructuring of the results and figures to be compatible with a non-specialist audience. It is written essentially as a structure report that relies heavily on supplementary figures for understanding and lacks an appropriate discussion of mechanisms and implications. The structural results are greatly underexploited in terms of analysis and should be put into the bigger picture of PG regulation during cell division.

Below I will point out my concerns and suggestions:

Some general points:

- The authors should review the description of the results and corresponding Figures to make the text amenable to a non-structural biologist audience.

We have modified the text to make it clear to general audiences.

- Most of the text cannot be understood unless you work through the supplementary Figures (there are 19). Please try to coordinate main text and figures in a narrative that is comprehensible for the reader.

Thank you for the suggestion. We have rearranged the figures in the revised version. Specifically, we have incorporated important findings from supplementary figures into the main figures to improve the overall flow and coherence of the story.

- Results should not be numbered and subdivided into small paragraphs, it makes the reading difficult.

Agreed. The numbers have been removed, thanks!

- The colors of many of the figures are very poor in contrast and make it impossible to interpret them correctly, in particular the main figures. Please use contrasting colors and check that the message described in the text corresponds to what is represented in the Figure.

Thanks for the suggestion. We have updated the figures with more contrasting colors.

- The Figure legends hardly describe anything. They should include the full information required to understand the figure.

Thank you for the suggestion. The figure legends have been modified accordingly by including more detailed information.

- Authors should analyse conservation patterns and discuss these, both with regard to the FtsEX complex that is present all over bacteria and the Rip family of hydrolases from *Corynebacterineae*, in particular the similarities and differences between RipA and RipC where now there are full length structures from each family available.

Thank you for the suggestion. The revised version includes a detailed comparison and new figures of the previously reported FtsEX systems as well as of other ABC transporters from family VII. A structural comparison of RipC and its structural homologues (RipA, RipB and Cg1735) is now provided (**see new Fig. S7**) in which we detail the active sites for these enzymes. We show that RipC presents the classical catalytic triad (Cys, His, His) observed in most of NlpC/P60 but different from that of RipA, RipB or Cg1735 which present a difference in the third residue (Cys, His, Glu). It has been reported that mutation of this residue Glu (E444 in RipA) by Ala results in the inactivation of the enzyme by inversion of the configuration of the catalytic Cys residue (Squeglia et al (2014)). Thus, in RipA, and likely in all RipA homologs, the Glu residue would act as a catalysis regulator by restraining its catalytic site's local flexibility. Interestingly, the conformational flexibility of the loop containing Glu444 is

limited due to the presence of a salt bridge formed between Glu444 and Arg458; a salt bridge conserved in the RipA homologs (**Fig. S7C, E, G**). This salt-bridge network is not observed in RipC (**Fig. S7 A**) presenting a Ser residue (S376) located in a small alpha helix not observed in any of the RipA homologs. Other relevant differences between RipC and RipA homologs are also observed in the loops around the catalytic center (**Fig. S7 I**) that result in a different shape and dimensions for the active binding site (**Fig. S7 J**). A summary of this information, due to space limitations, is now included in the revised version of the manuscript (**L428-L433, L437-L684**).

Specific points:

- Line 47: remove “for the first time”. Although this is the first full complex of the system there have been other studies providing partial molecular insights into activation mechanisms (ie Mavrici et al)

As suggested, we have eliminated a similar statements and instead included a comprehensive comparative analysis, specifically focusing on the RipA study.

- I am not sure it is necessary to make statements about specific design of novel antibiotics, as it is not clear to what extent RipC is essential for mycobacterial growth and division. In case the authors want to include this, they should discuss the literature on the conditional essentiality of RipC.

As suggested, these statements are removed.

Introduction:

- Rv numbers for FtsEX and RipC should be mentioned in the introduction

As suggested, we provided the gene identifier numbers for MtbFtsE (Rv3102c), MtbFtsX (Rv3101c), MtbRipC (Rv2190c/Cg2401) in the revised version.

- Line 65: “any known substrate” is this referring to the comparison with the ABC transporters? This sentence should be extended and clarified.

This sentence is now extended and clarified at **L78-L82**.

- Lines 69-72: this should be written more precisely, RipA does not interact with FtsEX but with SteB. However the second part of the sentence is correct in the sense that the auto-inhibitory CC is part of the hydrolase itself. Please rephrase

Thank you for pointing this out. The regulation of RipA (Rv1477/Cg1735) and RipC (Rv2190c/Cg2401) by FtsEX is still enigmatic, especially in Mtb.

Mavrici et al. showed that RipC (Rv2190c) interacts with the extracellular domain of FtsX (FtsX_{ECD}) but were unable to detect any interaction between RipA (Rv1477) and FtsX_{ECD} (However, this does not necessarily mean FtsX and RipA don't interact.

Negative data in biochemistry can have multiple interpretations, such as the reaction conditions, the use of a truncated protein, and the assay methods).

By contrast, Lim et al. identified the genetic relationship between RipA (Cg1735 but they called it RipC) and FtsEX by Tn-seq. $\Delta ripA$ and $\Delta ftsEX$ mutants phenocopy each other. They further demonstrated that RipA (Cg1735) interacts with FtsEX and SteAB (Fig. 10 of Lim et al.). Additionally, RipC' (Cg2402 and "RipA" in their manuscript) lacks the CC domain and unsurprisingly is in a different pathway.

Later, Gaday et al. elegantly clarified the confusion by performing a detailed phylogenetic analysis. They showed that the extracellular domain of SteB is sufficient to activate RipA (Cg1735) in vitro. This is likely done by derepression of the system through the CC domain.

Thus, we follow the naming system established by Gaday et al. We confirmed that RipC (Rv2190c) of Mtb forms a stable complex with FtsEX, but we could not exclude the possibility that RipA can also interact with FtsEX. Since the interaction between RipC (Cg2401) and FtsEX has never been tested in *C. glutamicum*, its role may be replaced by RipA (Cg1735) and SteAB. We have yet to work on the FtsEX complex of *C. glutamicum*, which may be a subject for future studies of other colleagues.

- Paragraph between lines 78-86: this paragraph should be rewritten to reflect more correctly what is known (or not known) in Corynebacterineae. For instance, not all species within this suborder have asymmetric cell division, proteins such as CwsA are not conserved in all. Also, I would not refer to "a few novel proteins" maybe species-specific proteins are more appropriate?

Rewritten as suggested (**L100-L111**), thanks!

- The introduction could do with more complete coverage of the mycobacterial PG remodeling literature.

Agreed. Focusing on the subject matter (FtsEX, RipC, and RipA), we added a few more relevant references and information about alternative RipA regulations to enrich the introduction. This topic is reviewed in PMID:31216697 and we decided to direct readers there for further information.

- Line 92-93, please add a reference to this statement.

Thanks for the indication. In the Results section, we have now included a detailed description of the inhibition modes in RipA and homologues (RipB, Cg1735) as well as in cell-division amidases accompanied with associated references. We have thus removed our previous statement (lines 92-93) from the Introduction section.

Results:

- The results need to be re-organised into a style compatible with an article of high impact. The numbering is not needed and the general organization needs to be revisited. For instance the first result is 1. Biochemical reconstitution and cryo-EM

study of FtsEX followed by 2. Cryo-EM structure of FtsEX followed by 2.1 Overall structure of FtsEX etc. All this should be one Result section on the structural characterization of FtsEX, starting from the purification, activity then the structure. Also there is a lot of technical information that precedes the actual finding, referring to supplementary data. This could be partially put into Materials and Methods. It makes the paper difficult to read and the important messages are drowned in the technical detail without a logical flow (for example lines 133-140).

Fully agreed. We have modified it accordingly to increase the readability and clarity of the results.

- Line 109: the D164A mutant is unable to bind ATP and the E165Q mutant is unable to hydrolyse ATP. Unless I missed this, I could not see any evidence on binding. Please clarify.

Apologies for the lack of clarity. The significance of these two mutants has already been well-established in the ABC transporter field and has also been tested in FtsE. We have addressed this concern by providing clarification and citing the relevant literature on FtsE at **L170-L173**.

- The authors should provide binding affinities for the FtsEX-RipC interaction in the presence of the different nucleotides or analogues. The affinities may (should) be affected (in particular with regard to what is later presented for activation and conformational changes upon hydrolysis).

As suggested, we performed pull-down experiments to compare the binding affinities of FtsEX-RipC interaction in the presence of ATP or its analogues. However, our findings revealed that the binding affinities for the FtsEX-RipC interaction are similar, regardless of the presence or absence of various nucleotides or analogues. Please refer to the updated **Fig. 1B** for details. This is actually consistent with our structural study, which demonstrates that the ECD domain of ATP-free MtbFtsEX is already well folded for RipC binding. Furthermore, our structures indicate that nucleotide binding does not induce significant conformational changes in the ECD domain.

- Lines 120-121: add standard deviations to values mentioned.

Added in **L183-L184**, thanks!

- Lines 123-125: it would be nice to make a more detailed description and interpretation of the results here, or even better in the discussion, together with how ATP hydrolysis affects activation. These sentences are not clear.

The interpretations have been carefully revised, and we have also provided an expanded explanation for the observed high basal ATPase activity of MtbFtsEX in the discussion section (**L1017-1053**).

- Result 2.2 on the unusual ECD conformation (lines 147-179): these important paragraphs describe conformational changes and structural comparisons that are difficult to understand without very clear and well described figures. ALL the figures referred to here are supplementary figures (S5-8) except for Figure 1G that does not exist! The authors need to choose the correct figures to properly illustrate their message in the main Figures and use the supplementary for additional info that is not essential for understanding the main message. For instance, Figure S5 needs to be a main figure as it actually shows the structural detail in a comprehensive manner (except for the colors that are not contrasted enough, and the legend that needs to be improved).

We have made the requested modifications and rearranged the figures accordingly. Specifically, the modified Fig. S5 is now represented as **Fig. 1**, the modified Fig. S6 is now **Fig. S3**, and Fig. S7 and S8 have been combined into the new **Fig. 2**. Additionally, the missing Fig. 1G has been corrected. We greatly appreciate your valuable feedback, as it has significantly enhanced the readability and clarity of our manuscript.

- Figure 1 B: the gels seem to have been cut and put together, they should be separated by a blank space. Figure 1E: in the schematic all the abbreviations need to be explained in the legend; the colors used should be the same between the schematic and the structure model and be contrasted enough to be distinguished. Figure 1F see comment below for Figure S7.

Thank you for your comments. We have addressed them in the revised version, including the addition of extra conditions to **Fig. 1B** by rerunning the gel.

- Figure S7 needs to be represented differently to understand what the authors want to show (maybe cartoon with surface superposed (transparent) and side plus top view?)

As suggested, we have updated **Fig. S7** by providing both Front- and Top- views for better clarification with ECD orientation indicated using arrows. Recognizing the significance of this figure, we have moved it to **Fig. 2**, now represented as **Fig. 2A** and **Fig. 2B**.

- Results section 3. Same comments as above on reorganizing the flow and moving some of the technical details into the materials. Section 3.1: again, the authors start straight with a comparison when the result is the structure of the complex. This needs to be described first and then use this result to compare with what has been shown for EnvC. The first panel of the Figure 2A describing this section cannot be a structure that has previously been solved by another lab...

Thank you for your suggestion. We have made several improvements based on it. Firstly, we have relocated some of the technical details regarding the EM data process and model building to the materials section (**L1247-L1266**). Additionally, we have reorganized the flow of the results by first presenting the structure results of RipC, while the structural comparison previously presented in session 3.1 has been placed

afterwards in **L722-L728**. Furthermore, we have adjusted Fig. 2A (see new **Fig. 5A**) accordingly by presenting our results first.

- In Figure 2, the active site of the catalytic domain is not well represented and hardly visible, the same goes for the details shown in Figures 2C-F. Please make these representations easier to understand.

We have updated **Fig. 2** accordingly, which is now in **Fig. 3A** and **Fig. 4A-4E**.

- Line 200-201, this comparison is also done just above, please regroup all the comparison linked to this, and again this is a full paragraph of structural comparisons with only supplementary figures. Include the main comparisons in a comprehensive manner in the main paper.

Agreed. In the revised version, we have moved this passage together with the previously mentioned one after discussing the results of the structure (**Now in L730-L775**).

- Lines 228-229: “typically”: what does this mean? This should be explained more precisely and with the correct literature cited. If the authors refer to other RIP structures from Mtb, this should be described and compared in a figure. It is impossible to know for people who are not familiar with the details of these proteins what is being referred to here.

Thanks for the suggestion. As indicated before, a structural comparison of RipC and its structural homologues (RipA, RipB and Cg1735) is now provided (see new **Fig. S7**) in which we detail the active sites for these enzymes and refer to associated references.

- The interaction of the catalytic domain with the coiled-coil and alpha1 should be compared (supplementary figure) with the closely related RipA structure from *C. glutamicum* (Gaday et al) that shows an autoinhibitory mechanism through the coiled-coil. This has important implications for conserved regulatory mechanisms in this important family of PG hydrolases of Corynebacterineae

The referee is absolutely right, the new **Fig. S7** also includes a description of the different inhibition modes observed in RipC structural homologues (RipA, RipB and Cg1735). While active sites in RipA and RipB are occluded by N-terminal segments of RipA (residues 260-321) or RipB (residues 30-97) (Böth et al J Mol Biol. 2011); the active site of Cg1735 from *C. glutamicum* (Gaday et al PNAS 2022) presents an alpha helix from the coiled-coil region blocking its active site in a similar fashion to that observed in RipC. Interestingly, a similar Glu residue is making an H-bond with the catalytic Cys residue in Cg1735 (E69) and RipC (E55). This information is now included in the revised version (**L428-L433**).

- Lines 244-245: “the activation is easy by simply moving alpha1 away from NlpC/P60 domain”. This is not a very scientific description, please rephrase.

Rephrased (**L688-L690**), thanks!

- Figure 3E: Why is there less FtsX for some of the lanes? This needs to be normalized and quantified, because if there is less FtsX, there will also be less RipC...

Thank you for bringing this to our attention. We have carefully repeated the experiments and updated the results accordingly. Please refer to the new **Fig. 4E** for the updated results.

- Figures 3A-D, again are very difficult to “read”, the changing orientations are confusing, as well as the approximate color correspondence between the main model and the insets. Again, the lack of contrast in the colors makes it all difficult to interpret.

Figs. 3A-D have been revised, with improved contrasting colors and clearer labels. Please refer to the updated **Fig. 4A-D** and **Fig. 6A** for these modifications. Thank you!

- Line 270 and Supplementary Figure S15: “...mutating these residues had no discernable effect on ATP hydrolysis”. Are the authors referring to all the mutants shown in Figure S15? Indeed, in this figure the ATPase activities of the double mutant F110A/F113A and the triple mutant F61A/F110A/F113A look rather different. If these differences are not statistically relevant, this should be shown. Also, a negative control is missing in this figure.

The differences between double mutant F110A/F113A and the triple mutant F61A/F110A/F113A are not statistically relevant, as confirmed by repeating the experiment again with more replicates for each condition. We have updated **Fig. S15** with a negative control added. Please see the updated **Fig. S8**.

- Result section 4.2 Figure 3F and G: these figures are difficult to interpret, they should be redrawn so that readers can easily understand the message the authors want to convey. It is unclear for example whether the two structures in 3F are shown in the same orientation. In Fig 3G there are too many things in the left panel, which might be separated, showing for instance the coil in exactly the same position and the tow domains in their apo or bound conformation (but not superposed). The right panel is completely unclear.

All suggestions have been incorporated into the revised figure. Please review the updated **Fig. 6** for these modifications, thanks!

- Lines 286-289: What does this mean? Where does this come from? And in particular line 289: this sentence is incomprehensible.

We apologize if the description was not clear enough. The structure of ECD of Mtb is similar to other ECDs previously reported (*P. aeruginosa*, *E. coli*, or *S. pneumoniae*) but presents two unique features: a disulfide bridge C73-C78 and a short α -helix,

residues 64-68, both connecting the lower and the upper lobes of the *Mtb*ECD. We think that these two features are important to maintain the binding site open during the “Match and Fit” mechanism to recognize RipC. In the revised version the sentence has been rephrased and a new figure has been included (**Fig. 6C**) to further explain it.

- Section 5/5.1: PG activity test. The info in Figure S16 is essential information and needs to be included in the main text. It also lacks controls (buffer alone, positive control, FtsEX alone, FtsEX+RipC without vanadate). In particular, the FtsEX+RipC control is essential to support the author’s claim that the nucleotide-free state represents the auto-inhibited form of FtsEX-RipC. The legend should be extended to include minimal information to interpret the figure (such as ATP concentration used, reaction conditions, etc).

Thank you for your suggestion. We have repeated the experiments, including the necessary controls and additional conditions. As a result, the previous **Fig. S16** has been updated and is now shown as **Fig. 7A** and **Fig. 8A** in the main figures. Furthermore, we have updated the legend with the relevant and necessary information.

- Mavrici et al (2014) showed that the ftsX ECD alone is capable of activating RipC PG hydrolase activity. The authors should discuss this observation in view of their findings that ATP hydrolysis is required to activate the complex

The relevant discussion has been incorporated (**L1049-L1053**). We propose that the activation of RipC by the ECD domain alone could be attributed to the flexibility of both the ECD domain and RipC in solution. At relatively high concentrations, this could shift the equilibrium towards the activated ECD-RipC complex, even in the absence of FtsEX. This observation mimics our recently reported study in the *Pae*FtsEX system, where EnvC activated AmiB (DOI: 10.1073/pnas.2301897120, *PNAS*).

- Figure S18 is incomprehensible, I don’t see how the authors can claim conformational rearrangements with these figures, please redo. Moreover, the supposed conformational rearrangement may be an overinterpretation, as the observed differences might also be due to a lower occupancy compatible with partial detachment of the catalytic domain.

Fig. S18 has been updated and is now presented as **Fig. S12C** in the revised version. This figure clearly illustrates the disappearance of NlpC/P60 domain density in the Type 2 conformation.

In addition, thank you for raising the concern regarding some possible overinterpretations of the conformational rearrangement. To address this, we have taken the following steps:

1. We have resolved a new structure that captures the FtsEX-RipC complex in its pre-hydrolysis state (**L808-L820**, and **Fig. 7**). This structure closely resembles the Type 1

conformation and confirms that RipC is in an autoinhibited state, as evidenced by well-visualized density in a locked configuration.

2. We have conducted PG hydrolysis functional assays in a systematic manner (**Fig. 7A**, and **Fig. 8A**), including controls from a Gram-negative bacterium with a recently elucidated activation mechanism (DOI: 10.1073/pnas.2301897120, *PNAS*). Additionally, we have included more conditions, clearly demonstrating that it is ATP hydrolysis, not just ATP binding, that activates RipC. This ensures the presence of a conformation with activated RipC in our sample in the presence of ATP-Mg²⁺.

3. The presence and absence of the catalytic domain of RipC have been further carefully analyzed by 3D flexibility analysis in CryoSparrc (**Movie 1 & 2**), validating the disappearance of its density in the Type 2 conformation due to its high flexibility.

After these functional and structural studies, we believe we have sufficient evidence to propose that these conformational rearrangements are real that lead to the activation of RipC. We hope this clarifies the matter. Thank you!

- Figure 4D, ininterpretable, again, due to colour choices, too small, too crowded.

Modified with enhanced clarity, thanks! Please check the updated **Fig.8E**.

- Results 5.2, again the figures are used inappropriately and are not clear or do not reflect the main text.

We have modified **Fig. 4**, now identified as **Fig. 8** in the revised version. We have also corrected the Figure citations in the main text. Thanks!

- Line 358: there is no evidence that allows to talk about unfolded RipC.

To provide clarity, we have replaced the term "unfolded" with "catalytic domain - unlocked."

As described in detail above, we have compelling functional evidence from our PG hydrolysis assay, which clearly demonstrates that the catalytic domain of RipC is unlocked and capable of PG hydrolysis. Additionally, our structural data reveal the drastic conformational rearrangement of RipC for activation. Although the high flexibility of the activated RipC prevents clear EM density capture without the support of PG in vitro, the comprehensive function-structure analysis presented in this study should be sufficient to confirm the activation of RipC and allow us to propose the working model.

- Line 361: what are MctA and MctB? There is no reference or explanation.

Corrected. MctA (Rv1697) and MctB (Rv1698) (Mycobacterial copper transport protein A and B) are the old names for SteA and SteB. For consistency, we will use SteAB throughout the manuscript.

- The discussion lacks precision and references. The authors should include a discussion on the Rip family in Corynebacterineae, analyse the similarities and differences, mention the fact that in this suborder of actinobacteria the most important hydrolase for cell separation is the SteA/B-controlled hydrolase RipA and not the FtsEX-controlled RipC, etc

Other Rip family proteins in Corynebacterineae are extensively discussed and compared in Gaday et al. and thus we are referring our readers there (**L89-L90**).

Gaday et al. in their manuscript "proposed that Corynebacteriales have evolved at least two distinct regulatory mechanisms". One is the SteAB-controlled RipA and the other is the FtsEX-controlled RipC. This proposal, although attractive, has not been rigorously tested in multiple actinobacteria by experiments. Thus, we would like to avoid reiterating their viewpoint here because our data neither support nor reject their model.

The genetic evidence presented by Lim et al. and Maeda et al. (PMID:26713407) contradicts this model. Both papers indicate that RipA (Cg1735 or CgR_1596) is in the same pathway as FtsEX (and SteAB). Additionally, Lim et al. could not detect ethambutol sensitivity in the RipC (Cg2402) mutants of *C. glutamicum*.

Gaday et al. suggested the interaction tests between RipA (Cg1735) and FtsEX are invalid due to aggregation issues, which "may account for non-specific binding". However, in Fig. 10A of Lim et al. the polar signals of FtsEX and RipA are clearly stronger than the signals of SteAB and RipA. If we believe the interaction between SteAB and RipA is real, which is reproduced by Gaday et al., there is no reason to reject the positive results for FtsEX and RipA.

The biochemical evidence presented by Gaday et al. should not be used as evidence to exclude the possibility that RipA can also be regulated by FtsEX, as it only shows that the extracellular domain of SteB can activate RipA in vitro. Whether SteA is involved, and if SteAB works together with FtsEX to control RipA, remain open questions that warrant future studies by Lim et al. and Gaday et al (as suggested by Fig. 5 in Gaday et al.). Additionally, the function of RipC (Cg2402) in *C. glutamicum* has not been studied. Thus, it is premature to say FtsEX-RipC is unimportant. Many PG hydrolases are redundant and their phenotypes are often masked unless multiple mutations are introduced. These controversies and discrepancies are outside of the scope of this study and therefore we left our discussion "less precise" to ensure accuracy.

Reviewer #1 (Remarks to the Author):

The authors have addressed all reviewer comments and suggestions, even going to the extent of adding a new structure to this work. I believe the manuscript has improved in the areas that were needed and that is ready for publication.

Reviewer #2 (Remarks to the Author):

After reading the newer submission of this manuscript, I think this new version is greatly improved and has addressed all my previous concerns and reservations as well as the ones of the other reviewers. I have no more issue with this manuscript, and I think it is a valuable study that should be published with no more modification.

As a side note, I am not personally convinced that how RipC is positioned onto the FtsEX has any role into the snapping of the septum during division, but this has no influence on the quality of the manuscript, and it is only a speculation in the discussion, so I do not really have a problem with the authors suggesting this.

Reviewer #3 (Remarks to the Author):

Thank you for thoroughly addressing all the comments on the previous version, the manuscript is now much easier to read and the new figures are excellent and really help understanding the structural basis of the proposed mechanism. I have no further comments and think this will be a valuable contribution to the field.